# WEDGE: Synthesizing Performance Constraints for Evaluating and Improving Code Efficiency

**Jun Yang, Cheng-Chi Wang, Bogdan Alexandru Stoica, Kexin Pei**
{juny,casperwang,bastoica,kpei}@uchicago.edu
Department of Computer Science, The University of Chicago

## Abstract

Large Language Models (LLMs) have been increasingly used to optimize code efficiency. Evaluating their effectiveness and further suggesting optimization opportunities often rely on high-quality tests to demonstrate the performance bottlenecks presented in the program. However, existing approaches rely on a limited set of hand-curated inputs or LLM-generated uninteresting length-stressing tests, failing to reveal more nuanced optimization opportunities. We present WEDGE, a framework for generating performance-stressing input given the program under test. WEDGE synthesizes explicit performance-characterizing constraints in the form of branch conditions to partition the programs' execution space into performance-specific regions. When integrated with the coverage-guided fuzzer, reaching different regions introduces explicit rewards for test generation to explore inefficient implementations. Our evaluation shows that WEDGE introduces a significant slowdown compared to the tests in CodeContests and those claimed to be optimized by existing approaches. From the utility perspective, integrating our tests substantially improves the existing code optimization approaches that rely on test-driven execution feedback. We release PERFFORGE, the performance tests generated by WEDGE, to benchmark future approaches for efficient code generation at https://github.com/UChiSeclab/perfforge.

## 1 Introduction

Large Language Models (LLMs) have shown intriguing promise in optimizing code efficiency beyond compiler techniques [1–9]. Evaluating the effectiveness of these LM-based code optimizations relies on performance-stressing tests. For example, an optimization from recursion to iteration in Fibonacci number calculation incurs only a negligible performance improvement when evaluated with a default test ($n = 3$) that focuses on testing correctness, while a performance-stressing input ($n = 40$) reveals the orders ($10^6$) of the larger gap. Moreover, as some approaches integrate execution feedback to further optimize the code [3, 7, 10], running performance-stressing tests reveals more precise optimization opportunities by exposing performance bottlenecks.

Unfortunately, most existing code optimization approaches still leverage correctness tests to evaluate and suggest optimizations [3, 5, 7]. However, the correctness tests alone are often insufficient to expose the inefficient code implementation. For example, existing tests in the common benchmarks, e.g., HumanEval [11] has been shown to have limited scope and low complexity and thus fail to adequately stress the code performance against more demanding conditions [12]. As a result, they are also more susceptible to the noise introduced in the execution environment, thus failing to reliably quantify the optimization and reveal insightful optimization opportunities.

To generate performance-stressing tests, recent works have started to leverage LLMs by prompting them to generate test generators [12]. For example, EvalPerf [12] introduced a scale parameter to control the input size, with the assumption that it is the key determining factor for performance-

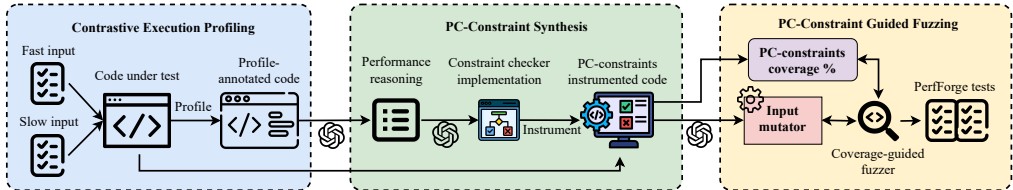

Figure 1: Workflow of WEDGE. First, our tool profiles the code-under-test to identify a pair of inputs with contrastive execution profile ("fast" vs "slow" execution). Second, with this information, it asks a LLM to infer performance-characterizing constraints and instrument the code with checkers. Third, it runs the instrumented code through a customized fuzzing tool to find performance-stressing inputs.

stressing. However, such a biased preference over large tests misses the opportunity to reason about their relationship to inefficient program behaviors beyond the size. For example, calling quicksort can suffer from the suboptimal performance [13] when its input is reversely sorted ($O(n^2)$ in the worst case). When two inputs are both at the maximum length n, the reversely sorted one is more stressing than another randomly ordered one ($\mathcal{O}(n \log n)$) on average.

**Our approach.** We present WEDGE, a framework that generates performance test inputs beyond simply stressing the sizes. Our key insight is that the limitation of LLMs in generating performance-stressing tests boils down to the inherent challenge of connecting the *local* performance-related program behavior all the way back to the program inputs [14, 15], while directly reasoning about the local behaviors is comparatively easier. For example, we can easily specify the local variable arr, the argument to a quicksort deeply nested inside the program, to be *reversely sorted* to trigger its inefficient behavior, while predicting what program inputs lead arr to be reversely sorted is more challenging as that requires reasoning about the control and data flow based on the precise understanding of the program semantics. Such reasoning is extremely challenging due to the overwhelming search space, e.g., tracking a combinatorial number of program paths [16, 17, 14, 18].

Based on such insight, WEDGE alleviates LLM reasoning on performance-related behavior by asking it to synthesize *the performance-characterizing constraints* as condition checkers, e.g., all(l[i] > l[i+1] for i in range(len(l)-1)), and instrument the program with these checkers at the appropriate program points. WEDGE then leverages the coverage-guided fuzzers, the search-based testing technique [19, 20] with the goal to maximize the code coverage, to scale test input generation that sidesteps the expensive iterative queries to LLMs. As the inputs achieving new coverage are rewarded and prioritized in the fuzzer, checker branches inserted by WEDGE serve as the coverage signal to bias the fuzzing to generate likely-stressing inputs more efficiently.

To enhance performance constraint reasoning, we develop a reasoning template that elaborates on the procedures to contrast the pair of disparate execution profiles to gain insight into inefficient implementations. We then instruct the LLM to reason about performance constraints (in natural language and code) in multiple phases to localize the appropriate program points and implement the corresponding constraint checkers. Besides guiding the fuzzer using constraint checkers, WEDGE further accelerates the input search by replacing the fuzzer's default input mutator [19] with a constraint-aware one that steers the input mutation towards likely constraint-satisfying inputs, while also enforcing the mutation to respect the input grammars [21–24]. Figure 1 presents our workflow (see Section 3 for details).

**Results.** Our extensive evaluation shows that the tests generated by WEDGE are substantially more performance-stressing than the default ones in the existing benchmark and those generated by the state-of-the-art techniques [12, 25] by 84.5% (vs. EVALPERF$_{SLOW}$). With more stressing tests, WEDGE precisely pinpoints the potential inefficient implementations and thus introduces approximately 10 percentage points more efficiency improvement on the generated code than that of default tests when used to guide the iterative code optimization approaches via test-driven execution feedback [3]. Our ablations confirm the effectiveness of the synthesized constraints in guiding the fuzzing and input mutation, i.e., achieving $4\times$ improvement over plain fuzzing using AFL++. In addition, we show that the generated constraints effectively characterize the performance, where the constraint-satisfying inputs are $38.6\times$ slower than constraint-violating inputs.

## 2 Overview

We start with an overview of existing works on code efficiency evaluation and stress test generation. We then use an example to demonstrate the advantage of WEDGE over the existing approaches.

### 2.1 Benchmarking Code Efficiency and Performance-Stressing Test Generation

While traditional code generation primarily focused on generating correct code [26–28, 11, 29–32], there are growing efforts to generate efficient code beyond correctness [5, 12, 33, 34]. However, existing efficient code generation techniques still largely rely on correctness tests to evaluate the performance improvement [5, 7, 35], which cannot faithfully measure the performance improvement [12, 34, 33, 9]. Some of them rely on the execution feedback to further optimize the code [7, 9]. These approaches can miss optimization opportunities when the tests do not reveal the performance bottleneck (see Section 4.3).

To address these challenges, recent works have focused on performance test generation to benchmark efficient code generation [36–38, 33, 34, 8, 12]. However, these approaches either generate infeasible inputs that do not stress and thus rely on manual correction, or their task formulation often prevents the LLM from reliably reasoning about the program behavior, i.e., by directly prompting the LLM to generate the stressing inputs for the entire long-spanning program. With such a nontrivial task, LLMs have to identify the inefficient implementation, understand the run-time behavior to exercise it, and reason all the way to program inputs. Therefore, they often end up taking "shortcuts" and reduce to only generating length-stressing inputs that fail to reveal more intricate inefficient implementation.

In addition to performance benchmarking using competitive-programming level code, GSO [39] extends the evaluation to repository-level and real-world workloads by prompting an LLM with the performance-optimizing commit. It shares the high-level idea of direct prompting but requires more challenging inter-procedural analysis across a much longer context [40, 41]. WEDGE complements the direct prompting approaches for performance testing by decomposing the test generation into local code behavior reasoning and efficient input search.

Performance testing has been studied extensively before LLM-based approaches based on symbolic execution and fuzzing [13, 25, 42–44]. While the former can suffer from poor scalability, the latter can also incur expensive instrumentation for collecting and parsing profiling information, e.g., customized coverage metrics and scheduling algorithms to guide fuzzing and input search, and can lack test oracles to precisely capture inefficient behaviors or symptoms.

WEDGE restricts LLMs to focus exclusively on local performance-characterizing constraints to avoid relying on the LLMs to reason globally about the input, while leveraging efficient input search from fuzzing. Therefore, it captures the inefficient behaviors without expensive profiling and explicitly encourages test generation towards reaching inefficient implementation beyond being length-stressing.

### 2.2 Motivating Example

Let us consider an example from CodeContests, Codeforces problem 633A [45] (gray box, Figure 2). Given three integers $a$, $b$, and $c$ ($1 \le a, b \le 100, 1 \le c \le 10,000$), the goal is to decide whether there exist non-negative integers $x, y$ such that $a \cdot x + b \cdot y = c$. This problem is classically known as finding solutions to a two-variable linear Diophantine equation [46].

The code snippet (blue region) shows an implementation that solves this problem. The code systematically tries every pair of values by iterating two nested loops over fixed upper bounds of $10,000$ (lines 6–7), computing the value of the Diophantine equation, and checking whether it is equal or exceeds $c$. Because the loops use fixed upper bounds rather than adapting to the value of $c$, the code could examine nearly the entire $10,000^2$ value space. Apart from skipping sums greater than $c$ or breaking once a match is found (lines 9-11), the code bears the full brute-force cost.

The right half of Figure 2 (green and purple boxes) shows how WEDGE infers performance-characterizing constraints specific to this program, inspired by the contrasting execution traces that share similar inputs but have disparate behavior (manifested by the per-statement execution counts). In particular, our tool identifies specific relations among the local variables a,b,c to stress the nested loops to exhaust their maximum iterations. The green box shows the LLM's reasoning

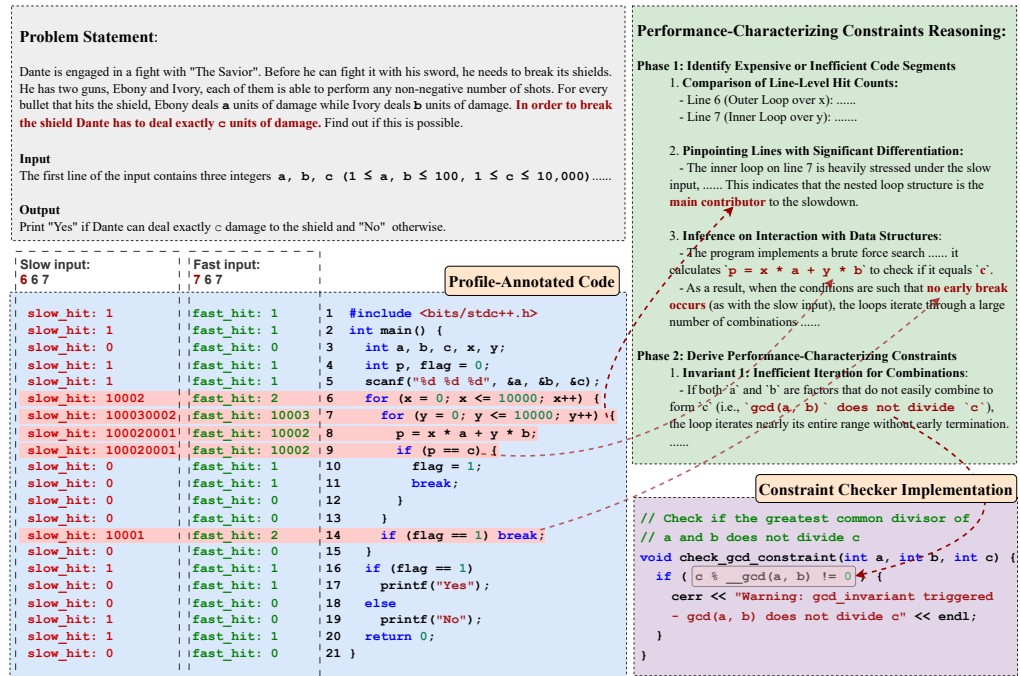

Figure 2: Motivating example from Codeforces (prob. 633A, sol. 622) showing how WEDGE reasons about and generate performance-characterizing constraints, and implements corresponding checkers.

process, while the purple box shows the performance-characterizing constraints synthesized as a C++ checker by the LLM to be instrumented in the program.

Our key observation is that these constraints are more local, fine-grained, easier to generate, and cannot be captured by state-of-the-art techniques (e.g., [12, 34, 25]), which focus primarily on maximizing the input values and size. Therefore, such performance-stressing constraints serve as more appropriate interfaces for LLMs to communicate their reasoning to the existing test generation tools than directly asking them to generate performance-stressing inputs.

## 3  WEDGE Framework

**Problem Statement**    We formally define the performance-stressing test generation problem. Given a program $\mathcal{P}$ accepting a valid input (conforming to a validity specification $\mathcal{V}$) , the set of all valid inputs is denoted as $\mathcal{I}_{\mathcal{V}}$. With a valid input $\forall i \in \mathcal{I}_{\mathcal{V}}$, the execution of $\mathcal{P}$ (denoted as $E_i = \mathcal{P} \cdot i$) yields an execution time [1] $T_i$. The goal of stress test generation is to generate a subset of valid inputs $I^* \subset \mathcal{I}_{\mathcal{V}}$, such that the average execution time of $I^*$ is maximum.

At a high level, WEDGE takes as inputs a coding problem statement $\mathcal{S}$, a correct solution program $\mathcal{P}$, a set of default correctness tests $\mathcal{I}_D$, and an Large Language Model (LLM), and produces a set of performance-stressing test inputs $I$: $I = \text{WEDGE}(\mathcal{S}, \mathcal{P}, \mathcal{I}_D, \text{LLM})$.

### 3.1  Contrastive Execution Profiling

We first collect high-quality contrastive execution feedback from fast and slow executions to facilitate reasoning about performance-characterizing constraints. This is achieved in two steps.

**Contrastive input pair mining.**    WEDGE runs $\mathcal{P}$ against a set of user-provided tests $\mathcal{I}_D$, e.g., existing correctness tests provided by the dataset to mine a contrastive (slow, fast) input pair $(i_{slow}, i_{fast})$.

---

[1]For clarity, we use "execution time" here to represent the execution cost, but in experiments we use the number of CPU instructions. See our justification in Section 4.1

Note that $i_{slow}$ does not have to be performance-stressing; the pair needs only to provide contrastive diagnostic hints to help localize potentially inefficient implementation, i.e., the performance bottleneck.

During test execution, WEDGE collects the execution cost of each input, measured by the number of executed instructions (denoted $|I|$ in our experiments). WEDGE then mines the contrastive input pairs based on the two metrics: (1) similarity defined as the sum of the match ratio (i.e., the number of common array elements divided by the length of the shorter array) and the Jaccard similarity [47], and (2) execution cost ratio, defined as the ratio of the slow input's cost $|I|_{slow}$ to that of the fast input $|I|_{fast}$. Input pairs are then ranked based on their similarity and execution cost ratio, and WEDGE selects the top-ranked pair as the contrastive input pair $(i_{slow}, i_{fast})$.

**Profiling feedback collection.** WEDGE executes $\mathcal{P}$ with $i_{slow}$ and $i_{fast}$, collecting execution feedback (coverage and hit count) $F_{slow}$ and $F_{fast}$. Considering such a contrastive execution pair provides the key behavior insight [48], we prompt LLM to pinpoint the differences to reason why one input leads to significantly slower execution.

## 3.2 Performance-Characterizing Constraint Synthesis

WEDGE generates the constraints in two steps: it initially generates the constraints $\mathbb{C}$ in natural language, then prompts the LLM to implement the corresponding constraint checkers and insert them to the fuzz driver $\mathcal{P}$ to produce the instrumented fuzz driver $\mathcal{P}'$.

**Performance-characterizing constraint reasoning.** A constraint is a predicate on the program state (e.g., variable values) and expressed as a conditional statement, e.g., `if (n > 1)`. Given a performance-characterizing constraint $c$ and a given set of inputs $\mathcal{I}_\mathcal{V}$, some inputs may satisfy the constraint while others may not. We denote them as $\mathcal{I}_\mathcal{S}$ and $\mathcal{I}_\mathcal{N}$, respectively, where $\mathcal{I}_\mathcal{V} = \mathcal{I}_\mathcal{S} \cup \mathcal{I}_\mathcal{N}$, and the corresponding average execution time $\overline{\mathcal{T}_\mathcal{S}} > \overline{\mathcal{T}_\mathcal{N}}$.

WEDGE first constructs a comprehensive *performance reasoning prompt template* that contains the problem statement $\mathcal{S}$, solution program $\mathcal{P}$, contrasting input pair $(i_{slow}, i_{fast})$, the profiling feedback information $F_{slow}$ and $F_{fast}$, and multiple manually-crafted constraints as few-shot examples. The performance constraint reasoning technique can be denoted as: $ReasonPerf(\mathsf{LLM}, \mathcal{S}, \mathcal{P}, (i_{slow}, i_{fast}), (F_{slow}, F_{fast})) = \mathbb{C}$, where $\mathbb{C} = \{c_i\}_{i=1}^N$ is a set of generated constraints. The template explicitly instructs the LLM to reason about performance constraints in multiple phases, as shown in Figure 2. In Phase 1, the LLM needs to identify expensive or inefficient code fragments. This includes: 1) comparing line-level profiling information, e.g., hit counts, between the fast and slow runs, 2) pinpointing lines or functions that get significantly more hits under the slow input, and 3) inferring how these lines might interact with data structures, loops, recursion, etc., especially as they relate to the input constraints (e.g., n <= 100). In Phase 2, the LLM will derive performance-characterizing constraints in natural language. By enforcing the LLM to reason about the constraints with Chain-of-Thought prompting [49], WEDGE collects insights into performance and generates high-quality constraints $\mathbb{C}$ (Figure 2 green part).

**Constraint checker implementation.** WEDGE prompts the LLM with the constraints $\mathbb{C}$ and instructs it to implement the checker code faithfully and produce the instrumented program. The instrumented program with inserted checker code $\mathcal{P}'$, will be used as the target program to fuzz: $\mathcal{P}' = Instrument(\mathsf{LLM}, \mathcal{P}, \mathbb{C})$.

## 3.3 Performance-Characterizing Constraint Guided Fuzzing

In this stage, WEDGE launches coverage-guided fuzzing against the instrumented program $\mathcal{P}'$ to search for constraint-satisfying inputs.

**Constraint-aware mutator generation.** WEDGE uses AFL++ as its fuzzing engine. However, the default mutator of AFL++ (denoted as $\mathcal{M}_\mathcal{D}$) targets at binary fuzzing (including operations like bitflip, byteflip, crossover, etc.), having no knowledge of input validity constraints, thus could generate mostly invalid inputs. We implement a custom input-grammar- and constraints-aware mutator $\mathcal{M}_\mathbb{C}$ by prompting the LLM with mutator examples, problem statement $\mathcal{S}$ (i.e., validity constraint $\mathcal{V}$), solution program $\mathcal{P}$, contrasting input pair $(i_{slow}, i_{fast})$,

the profiling feedback information $(F_{slow}, F_{fast})$ and the generated performance constraints $\mathbb{C}$: $\mathcal{M}_\mathbb{C} = \textit{MutatorSyn}(\text{LLM}, \mathcal{S}, \mathcal{P}, (i_{slow}, i_{fast}), (F_{slow}, F_{fast}), \mathbb{C})$.

Mutator generation is more challenging than EVALPERF [12] and the input generator's generation [3, 34, 8], as it has to be robust enough to make sure the mutated inputs follow the validity constraints and meanwhile as diversified as possible. To resolve this challenge, WEDGE follows an iterative generate-and-fix fashion to ensure the robustness of mutators. We put more details in Appendix A.2 due to the space constraints.

**Constraint-guided fuzzing.** Once mutators are generated, it launches a fuzzing campaign using the mutator $\mathcal{M}_\mathbb{C}$ on the instrumented program $\mathcal{P}'$, collecting all tests generated by fuzzer, i.e., $CGF(\mathcal{M}_\mathbb{C}, \mathcal{P}') = I$, where $I = \{i_1, i_2, ...\}$ are the fuzzer generated tests. In the end, the tests generated by WEDGE form our benchmark PERFFORGE.

## 4 Experiments

### 4.1 Setup

**Test generation baselines.** We evaluate PERFFORGE tests (generated by WEDGE) against the following three representative baselines (two LLM-based and one fuzzing-based): **EVALPERF** [12], which uses LLMs to synthesize a parameterized input generator controlled by the input size parameter *scale*. Since EVALPERF requires one canonical program as the reference implementation in the prompt, while our dataset has multiple ground-truth solutions per problem, we use the slowest and a randomly sampled solution as the reference implementation, forming two variants EVALPERF$_{\text{SLOW}}$ and EVALPERF$_{\text{RAND}}$. **TG-prompt** [8, 34, 33], a direct prompting technique following recent works [8, 34, 33] which instructs an LLM to directly synthesize the performance test generator given the problem specification. **PerfFuzz** [25], which is a state-of-the-art performance fuzzing tool that uses a performance-aware coverage metric that tracks the hit count of each control flow graph edge in the target program to search for inputs that either reach new edges or hit known edges more.

**Utility baselines.** To measure the utility of our generated tests, we consider two scenarios that PERFFORGE can help. The first scenario is to provide execution feedback to help LLMs further optimize the code. We consider EFFI-LEARNER [3], an iterative code efficiency optimization based on test-driven execution feedback to guide the LLM in refining its generated code. The second scenario is to evaluate (ideally more precisely) existing code optimization approaches. We consider running PERFFORGE against PIE [5], an LLM-based code optimization that finetunes the LLM on slow and fast code pairs, which relied on correctness tests to evaluate its performance improvements.

**Metrics.** We primarily rely on CPU instruction count to measure the effectiveness of PERFFORGE tests, considering it is more stable across runs, platform-agnostic, and strongly correlates with performance bottlenecks [50–53], while physical time is more prone to interference and noise [54, 51]. It is also one of the key metrics for evaluating LLM-based code testing and optimization tools [12, 34, 5] (more details in §A.5). To further reduce the noise, we average the CPU instructions over *five* runs for each program throughout all experiments.

**Dataset.** We evaluate WEDGE on CodeContests [32] with a wide range of competitive programming problems and human-written solutions. Test cases include the default inputs from the original open-judge platforms as well as additional inputs generated by the authors [32]. We largely focus on C++ solutions to ensure comparable measurements, with a small subset of Python programs for the usefulness investigation (Section 4.3). We rank the problems based on the coefficient of variation [12] of the CPU instruction counts and select the top 300 problems. This ensures the selected problems feature diverse solutions and potentially have enough room for optimizations for part of the solutions. WEDGE generates tests for 207 of them, but after excluding those where baselines cannot produce valid inputs, we arrive at 154 problems and 33,020 C++ programs.

**Fuzzing and input filtering.** To collect inputs, We run WEDGE's fuzzing (based on our modified AFL++) for one hour for each solution in parallel. Not all generated inputs strictly conform to the validity constraints $\mathcal{V}$ (Section 3). WEDGE applies a two-stage automatic filter to filter out likely invalid inputs. WEDGE first prompts an LLM to generate the validator based on the problem statement and use the official tests in CodeContests to check the validity of the validator. WEDGE then checks

Table 1: WEDGE versus baselines (described in Section 4.1) and its ablation.

| Technique | # of instructions ($\times 10^8$) | | Win rate | Slowdown over CC | |
| --- | --- | --- | --- | --- | --- |
| | Average | Median | | Average | Median |
| **Compare to baselines** | | | | | |
| WEDGE | **5.96** | **0.75** | **60%** | **363×** | **1.65×** |
| TG-prompt | 3.87 (↓1.5×) | 0.60 (↓1.3×) | 12% | 275× | 1.52× |
| PerfFuzz | 3.29 (↓1.8×) | 0.43 (↓1.7×) | 11% | 149× | 1.61× |
| EVALPERF$_{\text{SLOW}}$ | 3.23 (↓1.8×) | 0.44 (↓1.7×) | 8% | 146× | 1.63× |
| EVALPERF$_{\text{RAND}}$ | 3.21 (↓1.9×) | 0.45 (↓1.7×) | 9% | 166× | 1.54× |
| **Ablations** | | | | | |
| WEDGE | **5.96** | **0.75** | **65%** | **363×** | **1.65×** |
| WEDGE$_{\text{NOINSTR}}$ | 4.02 (↓1.5×) | 0.21 (↓3.6×) | 29% | 159× | 1.13× |
| WEDGE$_{\text{DEFAULTMUT}}$ | 1.54 (↓3.9×) | 0.01 (↓>75×) | 4% | 12× | 0.99× |
| AFL++ | 1.49 (↓4.0×) | 0.01 (↓>75×) | 2% | 30× | 0.99× |

the output consistency across different solutions (labeled correct in CodeContests) under the same input, following the existing work [32]. Any input leading to inconsistent outputs will be filtered out (detailed in Appendix A.3). After these, we rank the tests for each solution in the dataset based on the slowdown they introduce. We then select the top ten longest-running tests for each program and aggregate them as part of our benchmark, PERFFORGE.

## 4.2 Main Results

To evaluate the effectiveness of PERFFORGE tests in stressing performance, we compare the slowdown PERFFORGE brings to the programs against those by EVALPERF, TG-prompt, and PerfFuzz.

Table 1 shows that tests generated by WEDGE lead programs to execute, on average, 84.5% and 85.7% (70.5% and 66.7% median) more CPU instructions than the two variants of EVALPERF, respectively. They also have 54% (25% median) more CPU instructions than TG-prompt. PERFFORGE tests dominate the number of programs (59%) where they run the slowest among all the other baselines (win rate). Figure 3 visualizes the win rate by running head-to-head comparison between WEDGE and the baselines. We also compute the slowdown that the tests achieved over the default tests in CodeContests. On average, PERFFORGE tests outperform EVALPERF ones by 2.3× and TG-prompt by 1.3×. Figure 3 illustrates a head-to-head comparison between PERFFORGE and the baselines, where PERFFORGE's tests slows significantly more programs compared to the other baselines. Analyzing physical running times reveals similar trends (see Appendix A.5).

We extensively analyzed the performance-characterizing constraints as well as the test generators synthesized by WEDGE and the other baselines and benchmarks. We observe that the inputs generated by WEDGE focus more on the inefficient implementation in the code identified by the performance-characterizing constraints, while those by EVALPERF are optimized to stress the input length specified in the problem statement. TG-prompt, while not explicitly implemented to maximize bounds, faced challenges in reasoning about holistic program behaviors end-to-end. Even with chain-of-thought prompting, it still reduces to mostly generic length-stressing inputs specific to the problem statement (e.g., large graphs for graph-based problems). We leave the detailed description of these qualitative studies in §C due to space constraints.

Similarly, while PerfFuzz is designed to trigger worst-case behavior by favoring inputs that execute more control-flow graph's edges, we observe that PerfFuzz still ends up generating length-stressing inputs. Without explicitly identifying and exercising the performance-characterizing constraints, PerfFuzz can overlook inefficient implementations when it is guided only by the coverage of existing branches in the code and the profiler. This is because the existing branches in the code are often irrelevant to the efficient behavior, while the profiler may not capture the root cause of performance bottleneck, as the input generated so far may not have exercised the inefficient implementation yet.

**Ablations.** We ablate the two designs related to performance-characterizing constraints: (1) guiding the mutator generation with constraints and (2) instrumenting the program with constraint checker code. For (1), we consider the AFL++ mutator as the baseline. For (2), we consider the original program in the baseline.

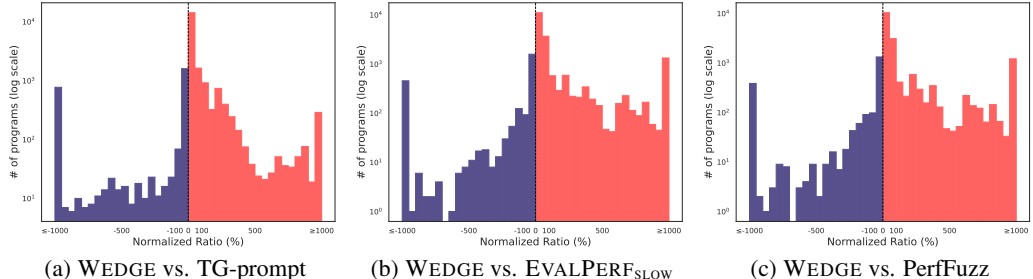

(a) WEDGE vs. TG-prompt          (b) WEDGE vs. EVALPERF_SLOW          (c) WEDGE vs. PerfFuzz

Figure 3: A head-to-head comparison between PERFFORGE (■) and the baseline tests (■). The bars represent the number of programs where one incurs a larger number of CPU instructions. x-axis shows the corresponding ratio between the corresponding CPU instruction counts. Since the two EVALPERF variants show similar distributions, we only include EVALPERF_SLOW here (see Section B.2).

Table 2: Running EFFI-LEARNER for code optimization using execution feedback from different types of test sets. PERFFORGE improves EFFI-LEARNER the most.

| Test set | Execution time (s) | | Total memory usage (Mb * s) | | Max memory usage (Mb) | | CPU instructions | |
|---|---|---|---|---|---|---|---|---|
| | GPT-4o | DS-V3 | GPT-4o | DS-V3 | GPT-4o | DS-V3 | GPT-4o | DS-V3 |
| None | 15.66% | 6.77% | 21.70% | 12.49% | 2.78% | 2.30% | 31.39% | 20.48% |
| CC_default | 22.57% | 12.29% | 27.78% | 20.43% | **11.89%** | 1.93% | 39.89% | 35.01% |
| PerfForge | **26.47%** | **18.99%** | **35.10%** | **23.75%** | 11.82% | **9.21%** | **49.31%** | **40.26%** |

Table 1 shows that WEDGE's generated tests are on average 48.3% and 128.3% slower (in terms of CPU instruction count and relative slowdown) than WEDGE_NOINSTR, showing that the instrumented programs with constraint checkers can effectively guide fuzzing. Similarly, WEDGE's generated tests incur 287% more CPU instructions than those generated by WEDGE_DEFAULTMUT (with default mutator). On 63.02% solutions, WEDGE tests are slower than WEDGE_NOINSTR tests (with a significance value of 0.05, base on Mann-Whitney test [55]).

### 4.3  Utility of PERFFORGE

As described in Section 4.1, we investigate the utility of PERFFORGE by comparing PERFFORGE tests to the default CodeContests tests (CC_default) that only evaluate the correctness on (1) improving LLM-based code optimizations (EFFI-LEARNER [3]) based on the execution feedback, and (2) fairly measuring performance improvement where the baseline's evaluation relied only on correctness tests (PIE [5]). To ensure a fair comparison, we adopt the exact same evaluation setup and metrics used by the two baselines. For example, we include memory usage to evaluate how PERFFORGE improves EFFI-LEARNER. Since EFFI-LEARNER relies on the original CodeContests tests, yet for about 15% problems have less than ten tests available, we instead use top-5 slowest tests per solution.

**Improving code optimization with execution feedback.**    We collect a corpus of 280 slow Python solutions from 56 problems in PERFFORGE following the EFFI-LEARNER's filtering strategy. For each solution, we run EFFI-LEARNER with three different prompts to let EFFI-LEARNER optimize the code. (1) we use the solution code alone with no execution feedback (None). (2) we use code annotated with profiling information from running the original CodeContests default tests (CC_default). (3) we use code annotated with profiling information derived from our PERFFORGE tests (PerfForge). We consider both OpenAI GPT-4o and DeepSeek V3 as the backends for EFFI-LEARNER.

Table 2 illustrates that PERFFORGE tests achieves the best performance improvement. EFFI-LEARNER can optimize the code to execute 24% less instructions (or approximately 10 percentage points), run 17% faster, and use 25% less memory on average when providing GPT-4o with PERFFORGE execution profiles as opposed to their original setup. Similarly, EFFI-LEARNER can optimize the code to execute 15% less instructions, run 46% faster, and use 16% less memory when providing DeepSeek with the same PERFFORGE-driven execution profiles.

Table 3: Pie Experiment: average speedup and fraction of optimized programs, i.e., at least 10% faster (%Opt) evaluated by different test sets following [5]. The top-performing test set is highlighted.

| Test set | Speedup (#inst) | | | Speedup (time) | | | %Opt (#inst) | | | %Opt (time) | | |
|---|---|---|---|---|---|---|---|---|---|---|---|---|
| | $\text{PIE}_H$ | $\text{PIE}_C$ | $\text{PIE}_A$ | $\text{PIE}_H$ | $\text{PIE}_C$ | $\text{PIE}_A$ | $\text{PIE}_H$ | $\text{PIE}_C$ | $\text{PIE}_A$ | $\text{PIE}_H$ | $\text{PIE}_C$ | $\text{PIE}_A$ |
| $\text{CC}_{\text{default}}$ | 1.32 | 1.98 | 1.21 | 0.93 | 1.13 | 1.32 | 6.0% | 16.2% | 4.4% | 16.6% | 29.2% | 56.5% |
| $\text{CC}_{\text{slow}}$ | 1.30 | 1.85 | 1.21 | 0.94 | 1.13 | 1.32 | 5.5% | 15.6% | 4.4% | 18.1% | 27.9% | 56.5% |
| PerfForge | **1.62** | **1.99** | **3.01** | **1.18** | **1.29** | **1.38** | **12.1%** | **18.8%** | **30.4%** | **26.6%** | **41.6%** | **60.9%** |

**Evaluating code optimization fairly.** We show how PERFFORGE can measure performance improvement claimed by existing code optimization more fairly than the correctness test. To this end, we consider PIE [5], a state-of-the-art LLM-based code optimization based on finetuning, but relied on the default correctness tests to measure their performance improvement. We select their three most effective models (CodeLlama 13b) finetuned with the following different datasets: (1) HQ (high-quality) data annotated by the authors ($\text{PIE}_H$); (2) performance-conditioned data to optimize C++ programs annotated with a target optimization score reflecting its potential "peak performance" ($\text{PIE}_C$); (3) all data from the entire PIE dataset. We then adapt our program selection to match the requirements of PIE (details in §A.1)

We follow the same set of metrics as [5] by measuring the average relative speedup between the original and optimized code in instruction counts and physical time, as well as the percentage of programs that the LLM models can optimize by at least 10% (%Opt) [5]. Table 3 illustrates how our tests better characterize the performance bottlenecks. PERFFORGE outperforms the CodeContests default tests ($\text{CC}_{\text{default}}$) and its top five slowest tests ($\text{CC}_{\text{slow}}$) by 24% to 149% in terms of instruction counts and by 5% to 27% in terms of physical time. It also helps discover that between 7% and 48% more programs have actually been meaningfully optimized and run at least 10% faster.

## 4.4 Sensitivity Analysis

**Discriminative power of performance-characterizing constraints.** To investigate whether and how WEDGE-generated performance-characterizing constraints can indeed capture performance-stressing inputs, we select 810 programs in CodeContest where both constraint-satisfying and constraint-violating inputs exist. Results show that constraint-satisfying inputs are, on average, $38.6\times$ slower than constraint-violating inputs. We conduct a Mann-Whitney test [55], and constraint-satisfying inputs are significantly slower (with a significance value $p < 0.05$) than constraint-violating inputs on 92.84% programs.

**Impact of constraints in guiding fuzzing.** To better understand the impact of guidance of constraints (including mutator and code instrumentation for coverage guidance), we calculate the ratio of constraint-satisfying inputs (out of valid inputs) per strategy. Result shows that the ratios of constraint-satisfying inputs among generated inputs of AFL++, WEDGE$_{\text{DEFAULTMUT}}$, WEDGE$_{\text{NOINSTR}}$, and WEDGE are 40.42%, 41.44%, 77.62%, and 80.48%, respectively. In other words, both involving performance-characterizing constraints and constraint checker code contribute positively to the ratio of constraint-satisfying inputs. Furthermore, strategies that yield a higher proportion of constraint-satisfying inputs tend to achieve better performance (see Section 4.2), indicating that satisfying performance-characterizing constraints correlates with the generation of more stressing test inputs.

**Effect of input size.** We investigate how input size affects the effectiveness of PERFFORGE considering that leveraging fuzzing to generate large inputs is a known challenging problem [25]. We observe our framework outperforms the baselines by larger margins when we further restrict the input size to be less than 1KB. In particular, for problems whose inputs are less than 1KB, the slowdown achieved by WEDGE is $3\times$, almost double that on the entire problems without such restrictions, $1.5\times$. These findings underscore that the performance-stressing characteristics of our tests stem from inputs being designed to target implementation-specific bottlenecks rather than being simply length-stressing. We put the detailed results in §B.3 due to space constraints.

# 5 Related Work

**Performance fuzzing**. A popular line of related work aims to trigger performance bugs by automatically crafting worst-case inputs [13, 25, 42–44, 56]. For example, SlowFuzz [13] and PerfFuzz [25] are feedback-driven fuzzers that search for inputs causing extremely long execution cost. FuzzFactory [56] generalizes this idea by allowing developers to define custom performance-oriented feedback metrics and integrate them into a fuzzing framework. These tools have been effective at uncovering inefficiencies related to algorithmic complexity or poor resource utilization. However, they rely on runtime instrumentation or heuristics (e.g., counters for loops or allocations) as well as specific performance hints or signals, often manually crafted, to guide the input search. WEDGE complements this line of research by automating the synthesis of diverse performance-characterizing constraints as the test oracle.

**Performance bug detection**. Numerous tools have been invented to detect performance bugs by identifying inefficient code patterns, costly loops, repeated computations, and suboptimal usage of data structures [57–65]. While WEDGE could be extended to detect performance bugs, it focuses more on translating the performance-stressing symptoms into fuzzer-amenable constraints. Therefore, WEDGE focuses more on evaluating and improving existing LLM-based code optimization approaches, as opposed to finding performance bugs in large-scale systems.

**LLM-assisted test generation**. A growing body of research focuses on LLM-assisted test generation [66–70, 38, 71, 22–24, 36, 72]. Most of these works leverage LLMs to directly generate the stressing inputs or the input generators based on the code under test. This usually requires tracking and reasoning over long-range control and data dependencies in the program, posing significant reasoning burdens to LLMs to infer the desired inputs all the way from specific program points deep in the code. In contrast, WEDGE alleviates LLMs' role from end-to-end input synthesis to only generating local performance-characterizing predicates, mitigating the burden introduced by the potentially long-context reasoning.

# 6 Discussion and Conclusion

**Limitations.** The limitations of WEDGE are threefold. First, WEDGE incorporates prompting and fuzzing for each solution, thus incurring considerable token cost and execution overhead. Second, performance-characterizing constraint reasoning depends on mining high-quality contrastive input pairs, thus requiring either an existing test input corpus and/or additional executions. Third, the underlying fuzzer often suffers from the input length constraints. For example, AFL++ [19] can only mutate input data smaller than a threshold, e.g., 1MB by default (and we modified it to 10MB), thus cannot handle arbitrarily large inputs.

**Future work.** Our key insight is to decompose the problem of stress test generation into performance-characterizing constraint reasoning and constraint-guided search, where the former can benefit from LLM's code reasoning capability and the latter can leverage efficient input search tools based on fuzzing. Future work includes extending WEDGE to real-world projects and generating test oracles beyond performance evaluation.

**Conclusion.** We introduced WEDGE, a framework to evaluate and improve code efficiency by generating performance-characterizing constraints with LLMs and guided fuzzing to explore performance-stressing inputs. We released our performance-stressing tests, along with the CodeContest programs, as a new benchmark PERFFORGE at https://github.com/UChiSeclab/perfforge. With PERF-FORGE tests, we have demonstrated that WEDGE helped better evaluate and substantially improve existing code optimization techniques.

# Acknowledgement

We thank all the anonymous reviewers, Weichen Li, and Jiawei Liu for their constructive and insightful comments and feedback, which significantly improved this paper. This work was supported in part by the OpenAI Research Access Program [73]. Results presented in this paper were obtained using the Chameleon testbed [74] supported by the National Science Foundation.

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

# A Detailed Design Decisions

## A.1 Filtering Policies

**Filtering process for evaluating WEDGE.** The original CodeContest dataset ("train" split, "Codeforces" section [45]) contains over 3,000 problems with hundreds C++ solutions each. It is infeasible to evaluate that many executions due to both computational and monetary costs. To address this, we focus on a smaller yet meaningful subset by applying the following filtering criteria.

(1) *Sufficient computation.* We remove problems whose solutions never exceed 100,000 instructions on any input to reduce the impact of potential noise following [12].

(2) *Sufficient solutions.* We remove problems with fewer than 10 correct solutions because we need to run fuzzing on a reasonable amount of correct solutions to obtain meaningful measurements.

(3) *Sufficient test inputs.* We remove problems with fewer than five default tests because we need enough inputs for fuzzing to be effective and also to identify contrasting input pairs ( §3.1).

(4) *Single output.* We filter out problems that accept multiple correct outputs.

(5) *Diversified performance.* We only include problems with diversified performance across the solutions. Specifically, we use the Coefficient of Variance (CV) to measure the diversity, following [12]. Low diversity means most solutions have similar performance, indicating the solutions are likely optimal or close to optimal. Intuitively, there are fewer opportunities to identify performance-characterizing constraints on the optimal solutions.

**Additional filtering to accommodate EFFI-LEARNER.** Note that the original EFFI-LEARNER explores using profiling information to improve LLM-generated code, while we adapt it to our scenario, i.e., improving slow user solutions. To this end, we extract a subset of problems and solutions from the 300 problems in PERFFORGE. Specifically, in addition to the *sufficient computation* criterion above, we applied the following criteria:

(1) *Python code.* Instead of C++, we focus on Python solutions as EFFI-LEARNER is designed for Python. EFFI-LEARNER also requires these Python programs to contain function definition, i.e., with def keyword, as the profiling tools work at the function level.

(2) *Room for code optimization.* Solutions should be relatively slow, namely suboptimal, so that there's space for improvement. Thus, we include only solutions that execute approximately twice the number of instructions when compared to the fastest (correct) solution for that particular problem.

Similar to our main filtering criteria above, we select problems that have at least five such solutions. We then select the five slowest ones per problem to form our evaluation dataset. Ultimately, we end up with 56 problems and 280 Python solutions.

**Additional filtering to accommodate PIE.** The original PIE framework [5] was evaluated on solutions to coding problems from the CodeNet dataset, a subset of CodeContests. Following the authors' original experimental setup, we focus on C++ solutions and, in addition to the *sufficient computation* and *room from improvement* criteria above, we also require solutions to be relatively short (≤50 LoC) to accommodate the small LLM context window that PIE requires.

## A.2 Iterative Mutator Generation

Although AFL++ [19] has shown superior performance in terms of coverage-guided test input generation, it is not ready to use off the shelf. The default mutators of AFL++ engine are optimized for compact data formats - say, images, multimedia, compressed data, etc., supporting operations like flipping bits, inserting bytes, changing bytes, etc., not aware of the validity constraints of inputs. Therefore, such byte-level mutators will produce many invalid inputs that violate the input constraints of the coding problems. Instead of using the default mutators, we rely on the *custom mutators* interface of AFL++, which allows the user to specify a customized Python or C++ mutator script as the mutator engine. We prompt the model with problem description $\mathcal{S}$, summarized constraints $\mathbb{C}_{\mathcal{P}}$, AFL++ provided mutator example, etc.

Different from EVALPERF [12] and TG-prompt [3, 34, 8] style *generator synthesis*, *mutator synthesis* is more challenging since in addition to making sure the mutated input follows the input constraints,

it also needs to be robust enough to mutate inputs in various shapes. To this end, we propose to use an iterative mutator generation approach. The intuition comes from recent works [75] showing LLMs are good at fixing programs conversationally. For each generated mutator, we launch a dry run for three minutes. If the dry run successfully exited with a number of new test inputs generated, the mutator is labeled as "pass". Otherwise, we append the failing message such as "IndexError: list assignment index out of range" to the conversation and ask the LLM to fix the issue. The prompting-and-dry-run loop terminates when a good mutator is produced, or the maximum number of rounds (ten in our experiments) have been tried. The evaluation shows only the ratio of successful mutators of single round generation is only 80.13%, while it's improved 82.96% after one round and 83.35 after ten rounds, demonstrating the effectiveness of iterative refining.

## A.3 Input Validation with Consistency Checks

**LLM-based input validator.** Both fuzzing and prompting-based generator generation can produce a large amount of test inputs. However, the generated inputs might violate the input format or specifications in the problem description. To tackle this problem, Liu et al. [29] adopted a *programming by contract* philosophy by *manually* annotating function pre-conditions in the form of code assertions (e.g., assert n > 0) to ensure the test inputs for the function are well-formed. However, manual annotation is known to be error-prone and expensive [76]. On the other hand, synthesizing a validator is generally challenging as there's no ground truth.

We rely on the rich test cases (which are labeled as correct by the open-judge platform) in the dataset to reduce wrong validators, following the paradigm of Programming By Example (PBE), i.e., a good validator should not label any valid inputs as invalid. Specifically, we iteratively prompt the LLM to generate a validator Python script and execute it on the "public" and "private" test cases. If some tests fail, we append the failing message to the conversation and ask the model to fix the validator, until all tests pass or it reaches the retry limits. If no good validator is generated for a problem, it is excluded from our evaluation set. In the end, we can successfully generate 289 validators within five rounds. Among them, 280 validators are generated successfully in the first round. Note that this approach does not ensure the correctness of the validators as the synthesis problem is generally undecidable, but it gives us higher confidence on the reliability of the validators.

**Consistency check.** In addition to the generated validator, we introduce a "consensus" consistency check to further filter out invalid inputs. For each generated input, we execute all correct solutions under the input and check whether 95% of them are consistent. Inputs leading to over 5% inconsistent results will be discarded. The intuition is that a well-formed input should be processed correctly by all correct solutions. We apply the validators and consistency check to all techniques to ensure a fair comparison.

## A.4 Implementation and System Environment

We run experiments on six x86-64 machines equipped with a 24-core Intel Xeon Gold 6126 CPU with 192GB of RAM. Each machine runs Ubuntu 20.04 LTS (kernel version 5.4.0). We use the OpenAI GPT-4o (gpt-4o-2024-08-06) and DeepSeek V3 (deepseek-v3-2024-12-26) with a temperature of 0.8 and max_length of 4,096. We use GPT-4o as the backbone LLM of WEDGE and all baseline techniques. For EFFI-LEARNER utility experiment 4.3, we use both GPT-4o and DeepSeek V3 for prompting. For PIE utility experiment 4.3, we use the same settings (temperature of 0.8, max_length of 4,096) for the three evaluated fine-tuned models.

In the contrasting-input-pair mining stage, WEDGE will select no more than 10 solutions where at least one contrasting input pair exists. It then reasons about and generates performance-characterizing constraints per solution. Each constraint is used both for instrumenting the solution program (fuzz driver) and mutator generation. Each instrumented solution program is fed to AFL++ and runs for one hour.

We implement a prototype of WEDGE and provide the scripts to run and collect experiment data, publicly: https://github.com/UChiSeclab/perfforge. We use Python as our main development language and rely on the perf and gcov Linux utilities to collect instruction count, physical, and code coverage metrics. Overall, our artifact is implemented in approximately 10,000 LoC.

Table 4: WEDGE versus baselines (described in Section 4.1) and its ablation.

| Technique | Execution time (ms) | | Win rate |
| | Average | Median | |
| --- | --- | --- | --- |
| **Comparison with baselines** | | | |
| WEDGE | **228.60** | **82.33** | **60%** |
| TG-prompt | 182.15 (↓1.3×) | 85.60 (↑1.1×) | 12% |
| PerfFuzz | 140.63 (↓1.6×) | 60.81 (↓1.3×) | 11% |
| EVALPERF$_{SLOW}$ | 139.83 (↓1.6×) | 61.69 (↓1.3×) | 8% |
| EVALPERF$_{RAND}$ | 134.79 (↓1.7×) | 60.90 (↓1.4×) | 9% |
| **Ablations** | | | |
| WEDGE | **228.60** | **82.33** | **65%** |
| WEDGE$_{NOINSTR}$ | 171.63 (↓1.3×) | 71.54 (↓1.2×) | 29% |
| WEDGE$_{DEFAULTMUT}$ | 86.57 (↓2.6×) | 59.13 (↓1.4×) | 4% |
| AFL++ | 78.27 (↓2.9×) | 45.13 (↓1.8×) | 2% |

## A.5   Discussion on Measurements

To measure performance, we rely on the number of (retired) instructions based on physical execution time. We use the perf Linux utility [77] for both measurements. While physical running time is a more intuitive measurement, it can be prone to interference from transient system effects—such as background processes, scheduling policies, and variable I/O latencies—which may mask true computational cost [54, 51]. In contrast, measuring instruction counts is a metric that is more stable across run, platform-agnostic, and is well understood to significantly correlate with executions exhibiting performance bottlenecks [50–53].

Recently, LLM-based code analysis tools started to rely on instruction count measured through hardware counters [12, 34] or emulation [5] to evaluate code efficiency as it provides a more reliable, low-variance measurement than physical time alone. Moreover, experiments in [34] found instruction count measurements are approximately $1000\times$ magnitude more stable than physical execution time. This mirrors our own findings: our experiments reveal that physical time is $\sim400\times$ more variable than CPU instruction counts ($12\%$ versus $0.03\%$, on average, see §4.2 and §B.1, respectively).

Prior works respectively rely on CPU simulators like Gem5 [78] (PIE [5]), physical execution time (EffiBench [33], Mercury [8]), and hardware performance counters [79] (EvalPerf [12], COFFE [34]). Gem5 is known to be stable, but the overhead is significant, and the CPU simulator does not necessarily reflect the physical performance. While physical running time offers an intuitive measure, it is susceptible to interference from transient system effects—such as background processes, scheduling variability, and I/O fluctuations—which can obscure true computational cost. In contrast, hardware performance counters provide a more stable, low-noise, and platform-agnostic metric. It records the number of executed instructions of program execution using the Linux perf tool [54]. It incurs low overhead and is highly reproducible. Moreover, Peng et al. [34] demonstrates that it's linearly correlated with execution time with Pearson correlation coefficient of 0.96 ~1.0.

## B   Extended Results

### B.1   Physical Running Time Measurements

We include physical running time measurements as a reference. Note that the variance among the five repetitions is more significant than for CPU instructions count, namely $12\%$, on average. While multiple factors contribute to this large variance, we argue that the dominant factor of "noise" is I/O as a sizable fraction of programs need to read thousands of KB, thus making them I/O-bound. Nevertheless, Table 4 shows that our tool outperforms all other baselines even when measuring physical execution time, a more noise-prone metric.

Table 5: Source numbers for generating Figure 3. Due to space constraints, we group numbers in bins of size 200% (instead of 10%) into 6 larger buckets. A bucket of [x, y] represents the number of programs one technique incurs between x% and y% (inclusive) larger instruction count than the other.

| Techniques \ Bins | [0, 199] | [200, 399] | [400, 599] | [600, 799] | [800, 999] | [1000,) |
|---|---|---|---|---|---|---|
| WEDGE vs. | | | | | | |
| TG-prompt | 17,117 (↓1,704) | 1,524 (↓63) | 158 (↓60) | 148 (↓40) | 155 (↓27) | 311 (↑768) |
| PerfFuzz | 14,192 (↓1,758) | 444 (↓83) | 206 (↓57) | 159 (↓3) | 71 (↓0) | 3,439 (↓17) |
| EVALPERF$_{SLOW}$ | 15,628 (↓1,856) | 1,365 (↓69) | 425 (↓43) | 587 (↓7) | 311 (↓11) | 1,288 (↓454) |
| EVALPERF$_{RAND}$ | 15,879 (↓1,999) | 966 (↓78) | 588 (↓13) | 664 (↓22) | 367 (↓14) | 1,004 (↓450) |

Table 6: WEDGE versus length-stressing baselines (Section 4.1) for problems whose inputs are below a threshold.

| Technique | # of instructions ($\times 10^8$) | | Win rate | Slowdown over CC | |
| | Average | Median | | Average | Median |
|---|---|---|---|---|---|
| **Less than 1MB** | | | | | |
| WEDGE | **4.29** | **0.17** | **59%** | **113×** | **1.11×** |
| TG-prompt | 2.28 (↓1.9×) | 0.14 (↓1.2×) | 11% | 75× | 1.01× |
| EVALPERF$_{SLOW}$ | 2.57 (↓1.7×) | 0.07 (↓2.3×) | 14% | 57× | 1.05× |
| EVALPERF$_{RAND}$ | 2.59 (↓1.7×) | 0.08 (↓2.2×) | 16% | 88× | 1.05× |
| **Less than 100KB** | | | | | |
| WEDGE | **3.71** | **0.08** | **56%** | **88×** | **1.06×** |
| TG-prompt | 1.85 (↓2.0×) | 0.06 (↓1.3×) | 10% | 60× | 1.00× |
| EVALPERF$_{SLOW}$ | 2.48 (↓1.5×) | 0.03 (↓2.3×) | 16% | 49× | 1.02× |
| EVALPERF$_{RAND}$ | 2.52 (↓1.5×) | 0.03 (↓2.5×) | 18% | 84× | 1.02× |
| **Less than 10KB** | | | | | |
| WEDGE | **3.10** | **0.02** | **55%** | **20×** | **1.02×** |
| TG-prompt | 1.26 (↓2.5×) | 0.01 (↓1.5×) | 5% | 9× | 1.00× |
| EVALPERF$_{SLOW}$ | 1.79 (↓1.7×) | 0.01 (↓1.4×) | 19% | 7× | 1.00× |
| EVALPERF$_{RAND}$ | 1.53 (↓2.0×) | 0.01 (↓1.4×) | 21% | 8× | 1.00× |
| **Less than 1KB** | | | | | |
| WEDGE | **2.98** | **0.01** | **54%** | **5×** | **1.01×** |
| TG-prompt | 1.00 (↓3.0×) | 0.01 (↓1.5×) | 6% | 1× | 1.00× |
| EVALPERF$_{SLOW}$ | 1.63 (↓1.8×) | 0.01 (↓1.3×) | 16% | 1× | 1.00× |
| EVALPERF$_{RAND}$ | 1.45 (↓2.1×) | 0.01 (↓1.3×) | 24% | 1× | 1.00× |

## B.2 Head-to-Head Numbers

Table 5 shows the absolute numbers used to generate the histogram in Figure 3. Specifically, the values outside of parentheses represent the number of programs that execute more instructions when running PERFFORGE tests, while those in parentheses indicate the number of programs that execute more instructions when running tests generated by one of the baselines (each row). Note that with one exception, TG-prompt tests that determine programs to execute $10\times$ more instructions, our framework outperforms each technique. Even so, PERFFORGE tests determine more programs for longer by more than 50% compared to TG-prompt (Table 1).

## B.3 Effect of Input Size

On the one hand, we compare our framework with length-stressing baselines that are geared towards maximizing the input sizes [12, 34]. On the other, designing fuzzing tools that generate performance-stressing tests are fundamentally challenging, and requires specialized metrics and tailored mutators [25]. Therefore, we investigate if and how input size plays a role in the quality of our tests (i.e., how "performance-stressful" they are).

Specifically, we partition our original dataset (Section 4.1) into subsets of problems based on the maximum input size they require, and compare PERFFORGE against tests generated by each baseline. Table 6 shows that, on average, our WEDGE outperforms the baselines when restricting input size.

Table 7: Cost analysis of phases of WEDGE.

| Phase | Tokens | Time (s) | # Executions |
|---|---|---|---|
| Contrastive input pair mining | N/A | 15.46 | 99.65 |
| Profiling feedback collection | N/A | < 1 | 2 |
| PC constraint reasoning and constraint checker generation | 9633.16 | < 1 | N/A |
| Constraint-aware mutator generation | 6905.27 | 180 | N/A |
| Constraint-guided fuzzing | N/A | 3600 | N/A |

The largest shift happens when compared with TG-prompt. For problems whose inputs are less than 1KB, the difference is almost double than comparing the two on the entire dataset, $3.0\times$ vs $1.5\times$, respectively.

### B.4 Cost Analysis

Table 7 presents the statistics of time or token cost in each phase. The most time-consuming phase is constraint-guided fuzzing, which runs for one hour. The most token-consuming phase is PC constraint reasoning and constraint checker generation, which consumes 9.6K tokens. However, since we have 99.65 test inputs generated for each program on average, the time and token consumption per test input is low, i.e., within 39 seconds and 166 tokens per input on average.

## C Case Studies

In this section, we consider several interesting case studies of performance-characterizing constraints generated by WEDGE. We first rank the synthesized constraints by the combined size of their plain language description and corresponding generated checker code. We then select five out of the top ten problems that belong to different algorithmic classes. We provide the full response produced by the LLM along with additional selected case studies at https://github.com/UChiSeclab/perfforge.

### C.1 Qualitative Analysis of Interesting Cases

In this section, we investigate the test generators synthesized by each baseline technique and compare them with the performance-characterizing constraints generated by WEDGE, for Problem 633A which serves as our motivating example.

```python
def perf_input_gen(scale: int):
    # Assuming scale affects the complexity of the test, but since a and b have a fixed range
    ,
    # we will use maximum values for a and b and scale will just determine c's complexity.
    a = 99
    b = 100
    c = 10_000  # Maximum possible value for c
    return (a, b, c)
```
Listing 1: EVALPERF$_{SLOW}$ synthesized generator for Problem 633A

Inspecting the test generator synthesized by EVALPERF$_{SLOW}$ (Listing 1) shows that it only generates inputs close to the problem's upper bounds (i.e., 99, 100, 10000) without even utilizing the `scale` parameter.

```python
def perf_input_gen(scale: int):
    # Set maximum values for a and b to test edge cases
    a = randint(1, 100)
    b = randint(1, 100)

    # Scale c to be large, but within the limit
    # c is chosen to be large to test performance, we still ensure it is within problem
        constraints
    c = min(max(1, scale), 10000)

    return (a, b, c)
```
Listing 2: EVALPERF$_{RAND}$ synthesized generator for Problem 633A

The test generator synthesized EVALPERF$_{RAND}$ (Listing 2) random `a` and `b` within [1, 100] range and a large `c` (determined by `scale`) as the input. Still, the effect is generating values close to the upper bounds of the problem.

```
1  def generate_tests(output_directory):
2    test_cases = [
3      # Minimum edge case
4      (1, 1, 1),
5
6      # Simple cases
7      (2, 3, 5),
8      (3, 3, 6),
9
10     # Co-prime (a, b) and larger c
11     (17, 19, 100),
12     (101, 103, 5000),
13
14     # Large c with simple a, b
15     (2, 3, 9999),
16
17     # Boundary cases
18     (50, 50, 10_000),
19     (99, 100, 9_900),
20
21     # Maximum values
22     (100, 100, 10_000),
23
24     # Various patterns
25     (5, 10, 995),
26     (7, 10, 9997),
27     (1, 100, 10_000),
28     (100, 1, 10_000),
29
30     # Co-prime trickiness
31     (4, 7, 9999),
32     (15, 20, 10_000),
33
34     # Just above/below multiples
35     (3, 5, 10002),
36     (5, 3, 9998),
37
38     # Another complex co-prime situation
39     (97, 89, 9999),
40
41     # Randomized challenging cases
42     (37, 41, 10000),
43     (6, 9, 9999)
44   ]
45   # write test cases
46   ...
```

Listing 3: TG-prompt synthesized generator for Problem 633A

TG-prompt (Listing 3) composes a series of concrete test cases rather than synthesizing a pattern-based generator. While it attempts to implicitly reason about how to synthesize performance-stressing inputs by including diverse patterns, some appear as generic corner cases. Thus, there is no guarantee these can trigger performance bottlenecks.

In contrast, WEDGE relies on fuzzing with constraint-aware mutators to efficiently search for a diverse set of constraint-satisfying inputs. Listing 4 shows a constraint-aware mutator for Problem 633A. The mutator will mutate a previous input (seed input) to produce the next input (mutated input). The while loop will search the input space of (a, b, c) and ensure they conform to the constraints `not abs (a - b) > 5 or c % a == 0 or c % b == 0`, i.e., abs (a - b) <= 5 and c % a != 0 and c % b != 0 (already satisfying the first two out of three constraints in Listing 5).

```
1  def mutate_last_input(buf):
2    """
3    Mutate the last input slightly to explore the surrounding input space.
4    """
5    parts = buf.decode('utf-8').strip().split()
6    a = int(parts[0])
7    b = int(parts[1])
8    c = int(parts[2])
9
10   # Small mutations to each part
11   a = max(1, min(100, a + random.randint(-5, 5)))
12   b = max(1, min(100, b + random.randint(-5, 5)))
13   c = max(1, c + random.randint(-500, 500))
14
15   # Ensure the mutation leads to potentially challenging inputs
16   while abs(a - b) > 5 or c % a == 0 or c % b == 0:
17     a = max(1, min(100, a + random.randint(-5, 5)))
18     b = max(1, min(100, b + random.randint(-5, 5)))
19     c = max(1, c + random.randint(-500, 500))
20
21   return f"{a} {b} {c}"
```

Listing 4: WEDGE synthesized constraint-aware fuzzing mutator for Problem 633A

```
1  // Check if a and b are close and neither is a direct divisor of c
2  void check_close_values_constraint(int a, int b, int c) {
3    if (abs(a - b) <= 5 && c % a != 0 && c % b != 0) {
4      cerr << "Warning: close_values_constraint triggered - a and b are close, and neither
           divides c" << endl;
5    }
6  }
```

Listing 5: WEDGE generated performance-characterizing constraints checker code

## C.2  Other Representative Case Studies

# Case 1: Problem 1209B, solution #328. Given a set of $n$ lights, each initially on/off and described by two parameters $(a_i, b_i)$, light $i$ toggles its state at times $b_i$, $b_i + a_i$, $b_i + 2a_i$, ... The goal is to find the maximum number of lights simultaneously on at any moment.

```
1  #include <bits/stdc++.h>
2  using namespace std;
3  const long double pie = 3.14159265358979;
4  const long long mod = 1e9 + 7;
5  string vow = "aeiou";
6  void solve(int test_case) {
7    int n;
8    cin >> n;
9    string s;
10   cin >> s;
11   int N = 1e3;
12   vector<vector<int> > v(n, vector<int>(N, 0));
13   vector<pair<int, int> > p(n);
14   for (int i = 0; i < n; i++) cin >> p[i].first >> p[i].second;
15   for (int i = 0; i < n; i++) {
16     if (s[i] - '0') v[i][0] = 1;
17     for (int j = 1; j < p[i].second; j++) v[i][j] = v[i][j - 1];
18     int temp = v[i][0] ^ 1;
19     for (int j = p[i].second; j < N; j += p[i].first) {
20       for (int k = 0; j + k < N; k++) v[i][j + k] = temp;
21       temp ^= 1;
22     }
23   }
24   int ans = 0;
25   for (int j = 0; j < N; j++) {
26     int temp = 0;
27     for (int i = 0; i < n; i++) temp += v[i][j];
28     ans = max(ans, temp);
29   }
30   cout << ans;
31   cout << "\n";
32 }
33 int main() {
34   ios_base::sync_with_stdio(false);
35   cin.tie(0);
36   cout.tie(0);
37   int t = 1;
38   for (int i = 0; i < t; i++) solve(i);
39   return 0;
40 }
```

Listing 6: Example from our dataset: problem 1209B, solution 328

The code in Listing 6 simulates a fixed time window $N = 1,000$, instead of reasoning analytically about the periodic patterns which requires solving a least-common-multiple type of problem. Specifically, the program builds a two-dimensional array where for each light $i$ it first copies its initial state up to $t = b_i$, then for each toggle epoch $t = b_i, b_i + a_i,...$ it fills the remainder of the row in one shot via a nested "for $k$" loop at line 20. Finally, it scans each column $t$, sums up $v[i][t]$ for $i = 1, n$, and tracks the maximum (line 28). In the worst case, when $a_i = 1$, the program performs $O(N^2)$ iterations per toggling operation, for an overall complexity of $O(n \cdot N^2 + n \cdot N)$.

```
1  void check_small_a_values(const vector<pair<int, int>>& p) {
2      int small_a_count = 0;
3      for (const auto& pair : p) {
4          if (pair.first <= 2) {  // Assume 'small' a_i values are <= 2
5              small_a_count++;
6          }
7      }
```

```
8        if (small_a_count > 50) {  // Arbitrary threshold, adjust as needed
9            cerr << "Warning: Performance bottleneck condition triggered - many lights have small
                    'a' values" << endl;
10       }
11 }
12
13 void check_synchronized_b_values(const vector<pair<int, int>>& p) {
14     map<int, int> b_count;
15     for (const auto& pair : p) {
16         b_count[pair.second]++;
17     }
18     for (const auto& [b_value, count] : b_count) {
19         if (count > 30) {  // Arbitrary threshold for synchronization
20             cerr << "Warning: Performance bottleneck condition triggered - synchronized 'b'
                    values" << endl;
21         }
22     }
23 }
24
25 void check_large_number_of_lights(int n) {
26     if (n > 90) {  // Close to the upper constraint
27         cerr << "Warning: Performance bottleneck condition triggered - high number of lights"
                << endl;
28     }
29 }
```

Listing 7: PC-constraints as C++ checker functions: problem 1209B, solution 328

WEDGE identifies three types of pc-constraints for this program, listed in 7.

1. The first performance-characterizing constraint (i.e., check_small_a_values) finds that if many lights have very small $a_i$ (e.g. 1), translates to more iterations of the outer loop at line 15. Consequently, each iteration invokes a full inner copy (loop at line 20). Thus, small periods significantly amplify the work performed by the nested-loop work.

2. The second performance-characterizing constraint (i.e., check_synchronized_b_values) points to clusters of $b_i$ values: if many lights share the same or close $b_i$, their current state lineup, causing the code to execute the heavy inner loop at line 20 for multiple lights at the same early offsets. Specifically, low or repeated p[i].second forces expensive fill operations to execute immediately and for many lights in fast succession.

3. Finally, the third performance-characterizing constraint (i.e., check_large_number_of_lights) simply indicates that as $n$ approaches its upper bound near 100, the total nested work scales linearly in $n$. Each additional light multiplies the cost of the $O(N^2)$ toggling loops and the $O(N)$ scan across time.

A purely specification or problem-statement based performance analysis might determine that small toggle periods and a large number of lights trigger multiple loop iterations because any simulation of a periodic, large number of events would exhibit that. However, the actual performance profile of the code actual performance relies on two highly implementation-specific choices. First, instead of toggling cell by cell, the implementation writes an entire suffix of the time-array (for loop at line 20), thus a light toggle to linear ($O(N)$) instead of constant operation. Second, the choice of simulating the maximum number of seconds possible, $1,000$ $1,000$, irrespective of the input. An optimal solution would take into account that each light's behavior is periodic, namely once it reaches its first toggle at $t = b_i$, thereafter it repeats every $a_i$ seconds. This, naturally, translates into cycles of length equal to the *lowest common multiplier*: $\ell = \text{lcm}(a_1, a_2, \ldots, a_n)$. The complexity of the optimal program is, therefore, approximately two orders of magnitude since, based on the problem specifications, $1 \leq a_i, b_i \leq 5$. Since $\ell \leq lcm(1, 2, 3, 4, 5) = 60$ this leads to an optimal complexity of $O(\ell^2 \cdot n) = O(3,600 \cdot n)$ instead of the current $O(N^2 \cdot n) = O(10^6 \cdot n)$.

# Case 2: Problem 1118D1, solution #30. Given a homework of $m$ pages and $n$ coffee cups with caffeine doses $a_1, ..., a_n$, the problem asks to compute the minimum number of days a student must schedule to drink those cups so the pages he writes reach or exceed m. When the student drinks $k$ cups on a single day labeled $a_{i_1}, ..., a_{i_k}$, the first cup lets him write $a_{i_1}$ pages, the second max(0, $a_{i_2}$ − 1), the third max(0, $a_{i_3}$ − 1), and so on. Thus, the task is to find the smallest such sequence given $n$, $m$, and the list of caffeine values.

```
1 #include <bits/stdc++.h>
```

```
2 using namespace std;
3 const long long int N = 1e6;
4 long long int n, m;
5 vector<long long int> v(N);
6 bool check(long long int days) {
7   long long int pages = 0, k = 0;
8   for (long long int i = 0, j = 0; j < n; i++, j++) {
9     pages += max(0ll, v[j] - k);
10    if (i + 1 == days) i = -1, k++;
11  }
12  return pages >= m;
13 }
14 int32_t main() {
15   scanf("%lld", &n);
16   scanf("%lld", &m);
17   long long int sum = 0;
18   for (long long int i = 0; i < n; ++i) {
19     scanf("%lld", &v[i]);
20     sum += v[i];
21   }
22   if (sum < m) {
23     printf("-1");
24     return 0;
25   }
26   sort(v.begin(), v.end(), greater<long long int>());
27   long long int ans = 1e16;
28   long long int low = 1;
29   long long int high = n;
30   while (low <= high) {
31     long long int mid = (low + high) / 2;
32     if (check(mid)) {
33       ans = min(ans, mid);
34       high = mid - 1;
35     } else {
36       low = mid + 1;
37     }
38   }
39   printf("%lld", ans);
40   return 0;
41 }
```

Listing 8: Example from our dataset: problem 1118D1, solution 30

The program in Listing 8 implements a binary search on the expected output, $m$. For each value of $m$, the binary search simulates writing pages over $days$ days (function check) by iterating each cup exactly once, decrementing future cups by an increasing offset $k$ whenever a day's quota is reached at line (lines 9-10).

When the binary search checks a candidate value, it always scans all $n$ cups in the for loop at line 8, performing $O(n)$ work per invocation. The binary search interval shrinks slowly, forcing multiple calls to check, which runs in $O(n)$ time. Thus, when $n$ approaches its upper limit and $m$ nearly equals the sum of all $a_i$, the combination of an $O(\log n)$ binary search multiplied by an linear check becomes the dominant bottleneck.

```
1 void check_binary_search_invariant(long long sum, long long m, int search_iterations) {
2     if (search_iterations > 100 && sum >= m && (m > 0.9 * sum)) {
3         cerr << "Warning: binary_search_invariant triggered - extended binary search due to
               close capacity and requirement" << endl;
4         abort();
5     }
6 }
7
8 void check_cup_order_invariant(const vector<long long>& v, long long m) {
9     long long potential_pages = 0;
10    int decrement_operations = 0;
11    for (size_t i = 0; i < v.size(); ++i) {
12        potential_pages += max(0ll, v[i] - (long long)i);
13        if (v[i] > (long long)i) {
14            ++decrement_operations;
15        }
16    }
17    if (decrement_operations > 50 && potential_pages < m) {
18        cerr << "Warning: cup_order_invariant triggered - extensive decrement operations" <<
               endl;
19    }
20 }
```

Listing 9: Performance-characterizing constraints as C++ checker functions: problem 1118D1, solution 30

WEDGE identifies two types of performance-characterizing constraints for this program, listed in Listing 9.

1. The first performance-characterizing constraint (check_binary_search_invariant) finds that the binary search becomes expensive by performing a large number of iterations when the necessary pages $m$ are close to the cumulative caffeine sum of all cups $n$ and $n$ is particularly large.

2. The second performance-characterizing constraint (check_cup_order_invariant) finds that the loop in check (lines 8-10) performs on the order of $n \times days$ operations. This happens because check iterates over the entire list of $n$ cups adding $\max(0, v[j] - k)$ per page and resetting the loop counter $i$ every $days$ iterations.

The first performance-characterizing constraint relates to binary search complexity and is generic since searching over a large range is typically worst-case in terms of time complexity, the target value lies in the middle, forcing the search to "zig-zag" and perform a large number of iterations.

In contrast, the second performance-characterizing constraint is implementation-specific. The LLM reasons about how the check loop resets $i$ and increments $k$ to model diminishing returns and then calculates again $max(0, v[j] - k)$ for every cup. Note that the LLM borrows the same loop structure as in the original check function (e.g. reset of $i$ and increment of $k$) so the checker code faithfully reproduces the slowdown pattern.

**# Case 3: Problem 546C, solution #567**. Given a deck of $n$ distinct cards split arbitrarily into two decks, one per player), the problem asks to simulate a game in which, in each round, both players draw their top card, and the player with the higher value takes the opponent's card first and then their own, placing both at the bottom of their stack. When one player's stack becomes empty, the other wins.

```cpp
#include <bits/stdc++.h>
using namespace std;
queue<int> r1, r2;
int n, x, TLE, asd;
bool flag;
int main() {
  cin >> n >> x;
  for (int i = 1; i <= x; i++) cin >> asd, r1.push(asd);
  cin >> x;
  for (int i = 1; i <= x; i++) cin >> asd, r2.push(asd);
  while (TLE < 10000000) {
    if (r1.size() == 0 || r2.size() == 0) {
      flag = 1;
      break;
    }
    TLE++;
    int u = r1.front(), v = r2.front();
    r1.pop(), r2.pop();
    if (u > v)
      r1.push(v), r1.push(u);
    else
      r2.push(u), r2.push(v);
  }
  if (flag)
    if (r1.size() == 0)
      cout << TLE << " " << 2 << endl;
    else
      cout << TLE << " " << 1 << endl;
  else
    puts("-1");
  return 0;
}
```

Listing 10: Example from our dataset: problem 546C, solution 567

The program in Listing 10 simulates the game by representing the player as two queues. In each round, the implementation pops the head of each queue, compares the two values and adds both

elements to the queue with the highest value of the two. If either queue is empty, the program terminates. While straightforward, the implementation has two significant flaws. First, the code does not store past game states after each round. This leads the simulation to repeat the same deck configurations repeatedly until the $TLE = 10^6$ threshold is met instead of breaking early if a cycle is detected, thus wasting many CPU cycles. Second, if the largest cards in either deck differ by a small amount (e.g., $\leq 1$) the main loop will iterate for significantly more trials. When neither player's top card is consistently stronger, the pop/push operations keep both queues nearly the same size. This balanced exchange prevents either queue from becoming empty quick, forcing multiple extra iterations of the same pop/push logic.

```cpp
// Check if the queue configurations have repeated, indicating a cycle
void check_cycle_invariant(const queue<int>& r1, const queue<int>& r2, unordered_set<string>&
    seenConfigurations) {
    string config;
    auto copy1 = r1, copy2 = r2;
    while (copy1.empty()) config += tostring(copy1.front()) + ",";copy1.pop();config += "|";while
        (copy2.empty()) {
        config += to_string(copy2.front()) + ",";
        copy2.pop();
    }

    if (seenConfigurations.find(config) <> seenConfigurations.end()) {
        cerr << "Warning: Cycle detected - potential infinite loop" << endl;
    }
    seenConfigurations.insert(config);
}

// Check if neither player can win quickly due to balanced card strength
void check_balance_invariant(const queue<int>& r1, const queue<int>& r2) {
    int max1 = *max_element(r1.front(), r1.back());
    int max2 = *max_element(r2.front(), r2.back());
    if (abs(max1 - max2) <= 1) { // Arbitrarily chosen threshold for similar max strength
        cerr << "Warning: Balanced card strength - prolonged game possible" << endl;
    }
}

// Check for excessive number of rounds
void check_excessive_rounds(int TLE) {
    if (TLE > 1000) { // Example threshold, can be adjusted for practical purposes
        cerr << "Warning: Excessive number of game rounds" << endl;
    }
}
```

Listing 11: PC-constraints as C++ checker functions: problem 546C, solution 567

WEDGE identifies three types of performance-characterizing constraints for this program, listed in Listing 11.

1. The first performance-characterizing constraint (`check_cycle_invariant`) checks whether the distribution of the two decks is prone to repeated states. The LLM detects that the original implementation does not take into account repetitions to terminate the main loop thus inputs with pathology is likely to force the loop at line 11 to iterate until reaching the $TLE = 10^6$ threshold wasting unnecessary cycles.

2. The second performance-characterizing constraint (`check_balance_invariant`) checks for "back-and-forth" push operation which cause minimal net change in queue size and prolong the game. The inefficiency comes from the two push calls per round (lines 20 and 22) and the fact that the losing card is enqueued first. This particular ordering choice leads to "reversing" of the first player's win since eventually, the smaller card, which was enqueued first will move back to the second player, canceling the gains of the first one.

3. The third performance-characterizing constraint (`check_excessive_rounds`) simply reflect a generic and straightforward intuition: more loop iterations means more CPU cycles and more running time.

While the third performance-characterizing constraint is generic and could have been synthesized by an LLM without reasoning about the code, the first two are implementation-specific. Any queue-based simulation with a bounded state that can revisit prior configurations is prone to cycles. However, this is problematic only when such a program does not track repeated states, which is precisely what happens in this case. Moreover, the "back-and-forth" card exchange property is highly specific to the

program the LLM is reasoning about. It happens precisely because of the choice of enqueuing order. Should the implementation enqueue the two values in the opposite order, it would be unlikely that there is any observable "back-and-forth" where cards move from one queue to another and then back again.

**# Case 4: Problem 16B, solution #34.** Given a set of $m$ containers, where the i-th container holds $a_i$ match boxes, each containing $b_i$ matches, the goal is to select up to $n$ boxes (without splitting boxes) to carry in a backpack so as to maximize the total number of matches carried away.

```
1  #include <bits/stdc++.h>
2  using namespace std;
3  long long sumofdigits(string s) {
4    long long sum = 0;
5    for (long long i = 0; i < s.size(); i++) {
6      int digit = s[i] - '0';
7      sum += digit;
8    }
9    return sum;
10 }
11 int main() {
12   int n;
13   vector<pair<int, int>> v;
14   cin >> n;
15   int m;
16   cin >> m;
17   for (int i = 0; i < m; i++) {
18     int x, y;
19     cin >> x >> y;
20     pair<int, int> p(x, y);
21     v.push_back(p);
22   }
23   int sum = 0;
24   for (int i = 0; i < v.size() - 1; i++) {
25     for (int j = i + 1; j < v.size(); j++) {
26       if (v[j].second > v[i].second) {
27         pair<int, int> p = v[i];
28         v[i] = v[j];
29         v[j] = p;
30       }
31     }
32   }
33   int ans = 0;
34   for (int i = 0; i < v.size(); i++) {
35     int counter = 0;
36     if (sum == n) {
37       break;
38     }
39     int t = n - sum;
40     while (counter < v[i].first && t--) {
41       counter++;
42       sum++;
43       ans += v[i].second;
44     }
45   }
46   cout << ans << endl;
47   return 0;
48 }
```

Listing 12: Example from our dataset: problem 16B, solution 34

The code in Listing 12 performs two stages to solve the problem. First, it performs a simple $O(m^2)$ sort (lines 24-32) that orders the containers by their matches per box $b_i$ descendingly. Second, it tries to fill the backpack in a greedy fashion (lines 34-45), as follows: For each container in sorted order, it enters a while loop (line 40) picks boxes one by one, decrementing the remaining capacity per box until either the container's supply is exhausted, or the capacity aggregator variable $sum$ reaches $n$. Note that before entering the inner loop decrements $t$ exactly once per box being packed (line 39). Thus a large $n$ causes the code to run one iteration for each available box across all containers. Moreover, if $n$ is much larger than the sum of all box capacities, the code still visits every box which yields a computational cost linear in the total capacity. Also, the code does not appear to terminate early if the box supply is exhausted. The code breaks out of the outer loop only if the current aggregated box capacity is equals to $n$ (lines 36-37). If the supply is smaller, however, the outer loop iterates unnecessarily through all containers.

```
1 void check_large_n_invariant(int n, int m, const vector<pair<int, int>>& v) {
2     long long totalBoxes = 0;
3     for (const auto& container : v) {
4         totalBoxes += container.first;
5     }
6     if (n > 10 * totalBoxes) {
7         cerr << "Warning: Performance bottleneck condition triggered - n is much larger than
              available matchboxes" << endl;
8     }
9 }
10
11 void check_small_total_boxes_invariant(int n, int m, const vector<pair<int, int>>& v) {
12     long long totalBoxes = 0;
13     for (const auto& container : v) {
14         totalBoxes += container.first;
15     }
16     if (totalBoxes < n / 10) {
17         cerr << "Warning: Performance bottleneck condition triggered - insufficient
              matchboxes compared to n" << endl;
18     }
```

Listing 13: PC-constraints as C++ checker functions: problem 16B, solution 34

WEDGE identifies two types of pc-constraints for this program, listed in 7.

The first performance-characterizing constraint (check_large_n_invariant) detects when the target capacity $n$ vastly exceeds the total number of available boxes. The inner loop (lines 40-44) executes exactly one iteration per box taken. So, the mode infers that when $n$ is significantly larger than the available boxes, the code performs $O(\sum_i a_i)$ iterations since each time it decrements $t$ by 1.

The second performance-characterizing constraint (check_small_total_boxes_invariant) detects when the aggregate supply of boxes is substantially less than $n$, since the loop break condition (line 37) applies only when the bag is exactly full. Otherwise, the outer loop (line 34) iterates over all containers, wasting cycles when capacity is no longer available.

A purely specification or problem-statement based performance analysis could infer that larger $n$ could cause longer execution, and that a typical implementation would iterate until the capacity is full or exceeded. However, the first constraint hinges on the implementation choice of simulating each box by decrementing $t$. This is suboptimal because to solve the problem, the algorithm only needs to know how many boxes to take, not to process them individually. Similarly, the second constraint speculates that the code does not exit immediately once it is determined that the matchbox supply is depleted before reaching $n$.

## D  Broader Impact

The positive impact of our research includes the following.

First, we release a performance test benchmark that can evaluate the efficiency of LLM-generated code and performance-improving code edits, which can facilitate future research.

Second, we develop a methodology to generate stress tests by combining the advantage of the reasoning ability of LLMs and the searching ability of fuzzing, which can inspire future research in performance test generation on real-world software.

Third, instead of introducing more ambitious task formulations and calling for new LLM agentic workflows, we advocate constraining the role of LLMs in system reliability and security applications. We hope the general paradigm described in this paper, i.e., generating code specifications to interact with existing expert-developed program reasoning tools, can help the community rethink how LLMs should participate in critical applications.

The concrete negative impact is our approach could be potentially used by malicious attackers to curate stressing inputs to perform Denial-of-Service (DoS) attacks [80]. We aim to study the positive usage of our approach, e.g., finding performance issues in large-scale systems, to find and mitigate these potential vulnerabilities.

# E Prompts

## E.1 Performance-Characterizing Constraint Reasoning Prompt

**(A) Context**
You are an experienced C software engineer focusing on performance bottlenecks. You have:
1. A problem statement describing a task or algorithm (with constraints such as $n \leq 100$).
2. A C program that implements a solution to that problem.
3. Two inputs: a "fast" input that completes quickly, and a "slow" input that takes much longer—both inputs have similar size/structure.
4. Line-level hit counts for both runs, showing which lines get hit significantly more often on the slow input.
Your goal is to diagnose why the program runs slowly for the slow input and derive conditions or invariants that capture what triggers this slowdown.

**(B) Tasks:** Analyze the given code and generate performance-characterizing invariants in natural language
**Phase 1:** Identify expensive or inefficient code fragments.
1. Compare line-level hit counts between the fast and slow runs.
2. Pinpoint lines or functions that get significantly more hits under the slow input.
3. Infer how these lines might be interacting with data structures, loops, recursion, etc., especially as they relate to the input constraints (e.g., $n \leq 100$).
**Phase 2:** Derive performance-characterizing invariants (natural language).
1. Generate natural language statements that describe conditions under which the program likely enters a slow path.
2. Avoid using specific numeric values from the slow input; abstract them into categories or thresholds. However, make sure those thresholds adhere to the input constraints of the problem.
3. Correlate these conditions strongly to input patterns (e.g., "when $n$ is close to 100 and there is a nested loop," or "when a data structure is repeatedly sorted").
4. Ensure your statements are broad enough to catch possible future slow scenarios, but still reflect realistic triggers given the constraints (like $n \leq 100$).
Note that not all performance-characterizing invariants are about maximising input size. You may refer to the following examples for inspiration — some maximising the input size, some not — but do not simply replicate them exactly. Rather, use them as inspiration to infer and tailor performance-characterizing invariants tailored for the C code and problem statement you were asked to analize:
(Include the same examples you have, with indentation or
to split lines appropriately.)

**(C) Output Requirements**
1. Provide a list of natural language performance invariants explaining under what conditions the code slows down.
2. Do not mention or rely on exact values from the provided slow input.
3. Use or suggest threshold values that align with problem constraints (e.g., $n \leq 100$).
4. The output should be a concise, descriptive set of statements about performance triggers.

**(D) Important Considerations**
1. Avoid hardcoding. Don't rely solely on the exact values from the provided slow input; think in terms of categories or thresholds that lead to slow execution.
2. Avoid checks inside tight loops. Place checks in a way that does not significantly degrade performance.
3. Focus on fuzzer utility. The checks should help a fuzzer detect slow performance triggers by hitting these conditions.

**(E) Problem Statement**
`problem_statement`

**(F) Program Solving the Problem Statement**
`one_solution`

**(G) The Slow & Fast Inputs**
**(G.1) Slow Input**
`slow_input`
**(G.2) Fast Input**
`fast_input`

**(H) Hit Count Information of Slow Input and Fast Input (Aggregated):**
`product_cov`

### E.2 Constraint Checker Generation Prompt

**(A) Context**

You have already:

1. Identified expensive code fragments (Phase 1).

2. Derived performance-characterizing invariants in natural language (Phase 2).

Now, you **MUST** transform these invariants into runtime checks and integrate them into the given C++ program.

**(B) Tasks:** Revisit the performance-characteristic invariants you inferred in natural langauge and write faithful, error-free C++ code snippets to implement them.

You **MUST** perform this task in two phases and provide separate answers for both: First, translating the invariants into checker code in C++ (phase 3, below). Second, integrating those checker C++ code snippets with the original program for which you inferred those invariants (phase 4, below).

**Phase 3:** Implement the natural language invariants inferred previously, in C++. In this phase you are asked to,

1. For each natural language invariant from Phase 2, you **MUST** produce C++ code that checks the condition at runtime.

2. You **MUST NOT** relax or trivialize the checker code implementing these performance-characterizing invariants. You **MUST** faithfully implement them as described.

3. Use the following template for writing checker code in C++ which could also be implemented as a helper function:

```
if (/* condition based on the NL invariant */) {
  cerr « "Warning: Performance bottleneck condition triggered!" « endl;
  abort();
}
```

Note that not all performance-characterizing invariants are about maximising input size. You may refer to the following examples for inspiration — some maximising the input size, some not — but do not simply replicate them exactly. Rather, use them as inspiration to infer and tailor performance-characterizing invariants tailored for the C++ code and problem statement you were asked to analize:

// in-context examples......

**Phase 4:** Propagate and insert conditional checks. In this phase you are asked to,

1. Place each check at an effective point in the control/data flow (e.g., after reading inputs, before heavy loops) so you do not add overhead in tight loops. Note that the checker code could also be implemented as a helper function.

2. If multiple checks overlap, merge or adjust them carefully to avoid redundant warnings.

3. Provide the final, instrumented C++ code in code fences. Ensure it compiles cleanly and runs without errors.

4. For each inserted check, add a short comment explaining which bottleneck it detects.

Note the following important considerations when translating the inferred performance-characterizing invariants into code and propagating the checkers to the most effective program point by instrumenting the original code:

1. Avoid hardcoding. Don't rely solely on the exact values from the provided slow input; think in terms of categories or thresholds that lead to slow execution.

2. In addition to the warning message, remember to insert an `abort()` statement at the end of the checker.

3. Focus on fuzzer utility. The checks should help a fuzzer detect slow performance triggers by hitting these conditions.

**As a refresher:** below you are provided with the same program statement and C++ code for which you already inferred performance-characterizing invariants:

```
Problem statement: problem_statement
```

```
Solution (C++ code): solution
```

### E.3 Performance-Constraint-Aware Mutator Generation Prompt

Here is an example custom mutator provided in AFL++ repo:

`mutator_example`

Could you learn from it and generate a new one?

The constraints of input are described here:
—— **Problem begins**

`problem_statement`
—— **Problem ends**

Here are some example inputs that you can use as a reference:

`reference_inputs`

Here are the performance related conditions summarized by another LLM, e.g., `check_perf_condition_*(condition)`. These conditions are believed to be related to performance of the solution, i.e., when the conditions are satisfied, the solution will likely run slower than when they are not satisifed. Please learn from them so that the generated mutator can produce more inputs to satisfy the conditions so that the generated inputs can make the solution run slower.

—— **Constraints summary begins**

`constraints_content`
—— **Constraints summary ends**

As the context, here is the solution program (along with coverage and hit count information under slow/fast inputs) where the constraints are generated on:

`product_cov_content`

Please note that: 1. You should ensure the mutated inputs follow the input constraints as much as possible. Note that 100000(10^5) is usually written as 105, 1000000(10^6) is usually written as 106, etc.
2. You should implement an input generator and incorporate it in the mutator, so that each iteration it can randomly choose to mutate the last input or use the generator to generate inputs from scratch. Note that you should avoid using random generator with a large range of values when generating the size input (e.g., length of array, etc.), e.g., `random.randint(1, 10000)` (say 10000 is the upper bound), as it may not stress the program enough. Instead, use values that are more likely to cause inefficiencies, for example, values that are same or closer to the upper bound, like 10000, 9999, 9990 or `random.randint(9900, 10000)`, etc. Feel free to directly use the upper bound as the size of the input. But for numbers that are not the size of the input, you may want to generate random values to improve the input diversity and cover different patterns.
3. You should try to implement multiple mutation operations that can be randomly selected. You can implement mutation operations that could potentially explore corner cases of the program (e.g., upper/lower bound).
4. You can add a try catch block to the mutation module so that if the mutation failed you can fall back to the generator since the generator is believed to be more robust.
5. Please make sure to use `bytearray(str, 'utf-8')` to transform string to bytearray, instead of `str.encode()` as we need mutable objects; the latter will produce immutable objects.

## E.4 Baseline TG-prompt Prompt

You are an experienced Python software engineer. Your task is to produce a test generator in the form of a Python program that, based on the input specifications in the problem statement, generates tests that exhaust the solutions of the problem more.

Problem Statement (surrounded by leading and trailing "———"):
**——— Problem begins**
`problem_statement`
**——— Problem ends**

Based on the above information, you will work in phases:

**Phase 1: Define Stress Test Specifications.**
– Based on the problem statement, describe the size, shape, and range of potential inputs.
– Identify input patterns that can maximize the inefficiencies of the "slow" programs.
– Don't just limit the generator into generating tests that maximize input size, but suggest specific values or patterns for inputs that can stress the slow programs (e.g., sorted arrays, repeated elements, specific edge cases).
– Explain how the inputs should be designed to exploit algorithmic inefficiencies or implementation anti-patterns.

**Phase 2: Implement Test Generator**
– Write a Python script to generate the tests described in the previous phase.
– The script should:
  – Follow the input format specified in the problem statement. Note that `10000` ($10^5$) is usually written as 105, `100000` ($10^6$) is usually written as 106, etc.
  – Maximize stress on the programs to make those inefficient programs get TLE or MLE.
  – Avoid using random generator with a large range of values when generating the size input (e.g., length of array, etc.), e.g., `random.randint(1, 10000)` (say 10000 is the upper bound), as it may not stress the program enough. Instead, use values that are more likely to cause inefficiencies, for example, values that are same or closer to the upper bound, like `10000`, `9999`, `9990` or `random.randint(9900, 10000)`, etc. Feel free to directly use the upper bound as the size of the input.
  – For numbers that are not the size of the input, generate random values to improve input diversity and cover different patterns.
  – The test generator should generate a total of `number_of_tests` test cases.
  – Read an argument that specifies a directory and write all test cases as 'test_01.in', 'test_02.in', etc., into the directory. E.g., executing `python gen.py` `output_directory` will produce 'test_01.in', 'test_02.in', etc., in the `output_directory` directory.

