# OpenReview forum: "WEDGE: Synthesizing Performance Constraints for Evaluating and Improving Code Efficiency"
_NeurIPS.cc/2025/Conference — NeurIPS 2025 poster_

### Official Review · Reviewer_zqXK · 2025-06-24

**Clarity:** 3
**Significance:** 2
**Originality:** 2
**Rating:** 4
**Confidence:** 5

**Summary:**

The authors propose a two-step pipeline for generating program inputs that cause slow performance. The first step uses an LLM to generate and instrument the target program with formulas that, when evaluated as true, should result in slow program execution. The second step uses an existing fuzzer to generate inputs that satisfy the formulas. The authors empirically evaluate their pipeline and find that, on average, they can generate program inputs that lead to longer execution times (in terms of the number of CPU instructions) than those generated by EVALPERF. Even better, they show that code optimizers can use their pipeline to generate more performant code.

**Questions:**

- What metrics or evidence did the authors use to determine that [12] is state-of-the-art in performance fuzzing?
- Can the authors provide citations or empirical evidence to back up their claims about related work? Many performance-fuzzing papers are not cited.
- What kind of performance bugs can the performance constraints capture? For example, can they express stateful performance issues? What other limitations are there? See, e.g., “SPIDER: Fuzzing for Stateful Performance Issues in the ONOS Software-Defined Network Controller” for an example of a test generator that does handle stateful performance issues.

**Ethical Concerns:**

["NO or VERY MINOR ethics concerns only"]

**Final Justification:**

My biggest concern with this work is how it engages with related work, both in
terms of the literature review and the empirical evaluation. During the
rebuttal, the authors provided new data to support their claims (e.g., that
their performance overhead is lower than existing tools) and made important
distinctions between kinds of related work (e.g., symbolic execution versus
fuzzing).

My second biggest concern was with over-claiming. The evaluation uses simple
programs and the authors do not outline the scope of their work (e.g., that it
cannot easily be extended to handle stateful programs). The authors acknowledge
this in their rebuttal.

Given the authors rebuttal, I am willing to raise my recommendation to "4:
Borderline accept". I worry that the changes the authors should make to the
final version of the paper are too big, but I do believe that the authors agree
with the changes that need to be made.

**Limitations:**

No. See question 3 above.

**Quality:**

2

**Strengths And Weaknesses:**

## Strengths
- Generating performance constraints to guide fuzzing is a nice idea.
- The paper is easy to read; the figures and tables are informative.

## Weaknesses
- Makes strong claims about related work without evidence to support them. For example, “However, they are known to suffer from scalability issues such as path explosion, fine-grained performance profiling overhead, and the lack of oracles to precisely capture inefficient behaviors or symptom” is a sweeping dismissal of a large and diverse body of related work that the authors present without evidence. See question 2 below.
- Questionable baselines for the empirical evaluation. See, e.g., question 1 below.
- Does not consider limitations of the fundamental contributions of the approach. See question 3 below.

---

> ### Author Rebuttal · Authors · 2025-07-31
>
> We appreciate the time and effort you took to review our paper and provide valuable comments that will help us improve it further.
>
> **Q1: Makes strong claims about related work. Many performance-fuzzing papers are not cited.**
>
> Thanks for pointing this out. We apologize for the extremely condensed discussion of performance-fuzzing-related works and the imprecise claim. Making a sweeping dismissal was absolutely not our intention, nor do we intend to make misleading claims. In fact, we sincerely respect the great line of literature on fuzzing and performance testing, which motivated our design and helped us distinguish Wedge from all existing LLM-based input generation approaches. While we do not attempt to make an excuse here, our original motivation was to benchmark existing LLM-based code optimization and compare it to recent LLM-enabled performance testing works. This unfortunately biased our citations to these recent LLM-based works, but we do not intend to claim they are the state-of-the-art. We have revised the paper to include an extensive discussion of performance-fuzzing-related works.
>
> Regarding “path explosion, fine-grained performance profiling overhead, and the lack of oracles to precisely capture inefficient behaviors or symptoms”, we totally agree that the claim is too strong and unfair. We apologize for such a misleading summary, as we tried to save space in our original submission. Please allow us to elaborate on our rationale in the following.
>
> Specifically, for path explosion, we refer to static approaches like symbolic execution, where the path explosion is a known problem [1, 2]. This is to actually argue the advantage of our fuzzing-based approach, but it got merged into the discussion of “traditional approaches”. We have fixed the description in the paper.
>
> For lack of oracles, we attempted to argue that some performance fuzzing does not have an explicit and reliable oracle, e.g., slowfuzz [3] relies on resource usage as guidance, but it may only serve as an approximate proxy to represent a performance issue. In fact, we admit the constraints generated by Wedge may not be perfect either, but complementary to this metric (see our new experiments below). We will tone down the claim.
>
> For profiling overhead, we add the following experiments to compare Wedge’s efficiency of the generated tests with those of Perffuzz [4], one of the well-known performance fuzzer. Specifically, we run perffuzz, perffuzz on Wedge’s instrumented programs (perffuzz_wedge_inst), Wedge (AFL++ with instrumented programs), and the original AFL that perffuzz is built on with Wedge’s instrumented programs (afl_wedge_inst), and compare the **average number of target program executions per second** (exec_per_sec).
>
> | Technique  | Average exec_per_sec |
> | -------- | ------- |
> | perffuzz |59.68    |
> | perffuzz_wedge_inst | 47.05     |
> | wedge   | 87.11    |
> | afl_wedge_inst | 114.69    |
>
> We can see that AFL with perffuzz’s profiling incurs significant overhead (perffuzz and perffuzz_wedge_inst), executing the fewest instructions per second. When turning off such profiling and using the base AFL that perffuzz built on with wedge’s instrumented program (afl_wedge_inst), the number of executions per second almost doubles.
>
> Importantly, we run these generated tests on the programs of the corresponding problems and find that the increased profiling cost does not lead to improved performance-stressing capabilities.
>
> | Technique                | perffuzz | perffuzz_wedge_inst | Wedge | AFL_wedge_inst |
> |--------------------------|----------|----------------------|--------|----------------|
> | # of instructions (×$10^8$) | 0.71     | 0.76                 | 1.15   | 0.77           |
> | Slowdown over CC         | 0.89     | 0.91                 | 1.08   | 0.92           |
>
> We observe that the original AFL with wedge’s instrumented programs can generate more stressing tests than tests generated by perffuzz and perffuzz with wedge’s instrumented code. Moreover, using the complete design in Wedge obtains the most stressing tests compared to all the settings.
>
> [1] Cadar, Cristian, Daniel Dunbar, and Dawson R. Engler. "Klee: unassisted and automatic generation of high-coverage tests for complex systems programs." In OSDI, vol. 8, pp. 209-224. 2008.
>
> [2] Chen, Bihuan, Yang Liu, and Wei Le. "Generating performance distributions via probabilistic symbolic execution." In Proceedings of the 38th International Conference on Software Engineering, pp. 49-60. 2016.
>
> [3] Petsios, Theofilos, Jason Zhao, Angelos D. Keromytis, and Suman Jana. "Slowfuzz: Automated domain-independent detection of algorithmic complexity vulnerabilities." In Proceedings of the 2017 ACM SIGSAC conference on computer and communications security, pp. 2155-2168. 2017.
>
> [4] Lemieux, Caroline, Rohan Padhye, Koushik Sen, and Dawn Song. "Perffuzz: Automatically generating pathological inputs." In Proceedings of the 27th ACM SIGSOFT international symposium on software testing and analysis, pp. 254-265. 2018.

---

> > ### Author Response · Authors · 2025-07-31
> > **Additional questions and responses**
> >
> > **Q2: What kind of performance bugs can the performance constraints capture?**
> >
> > Thanks for raising this point, which gives us another opportunity to provide further clarification. WEDGE is not a bug-finding tool, but rather aims to provide a suite of tests for benchmarking performance optimization. For example, we use only a subset of the existing solutions (see Section 4 and Appendix) to generate performance-stressing inputs, but our evaluation focuses on keeping only those inputs that incur the largest instruction counts across all programs on average for the specific problem. Our contrastive inputs only help localize potentially inefficient implementations, but may not be effective enough to identify a performance bug.
> >
> > Therefore, we do not attempt to overclaim that we are able to capture performance bugs. In fact, a good performance bug finding tool complements Wedge by providing a high-quality program point to generate performance-characterizing constraints, although with the caveats that the generated tests can potentially overfit the particular programs with the performance bug, but do not generalize to other programs. From this perspective, Wedge can be viewed as focusing more on generating proof-of-concept (PoC) inputs given the known bug (or potentially inefficient implementation) program point, where we employ additional vetting to ensure these inputs generalize to other programs for stress testing.
> >
> > **Q3: Can you express stateful performance issues? What other limitations are there? See, e.g., “SPIDER: Fuzzing for Stateful Performance Issues in the ONOS Software-Defined Network Controller” for an example of a test generator that does handle stateful performance issues.**
> >
> > Thanks for pointing us to this exciting work! Indeed, SPIDER introduces advanced features to handle stateful performance issues in event-driven SDN controllers, whereas Wedge does not currently support stateful fuzzing. It would be an interesting future work to generate not only performance-characterizing constraints, but also a sequence of temporal states that serve as preconditions to reach the generated states. In this case, the reasoning requirements posed on LLMs become more challenging, as the model must reason across the entire code base to determine dependencies and relationships between states and target performance constraints. We believe Wedge needs nontrivial augmentation to handle stateful performance fuzzing. Thanks for introducing this exciting direction for us to extend our idea.

---

> > ### Comment · Reviewer_zqXK · 2025-08-03
> >
> > Thank you for providing empirical evidence for your claims and for engaging
> > more thoroughly with related work. The new data, along with the context in your
> > answers to reviewer b8hS about cost in time and tokens, will improve the paper.
> >
> > Your answers to questions 2 and 3 are also helpful. These clarifications, and
> > the honest discussions about scope and limitations, will also help improve the
> > paper.

---

> > > ### Author Response · Authors · 2025-08-03
> > >
> > > Thank you for your engagement with our rebuttal and your positive feedback. Your comments are crucial in helping us strengthen the paper, and we are encouraged that you found the new additions valuable. For the updated version, we are working on integrating the new results on performance fuzzing baselines, fine-grained token cost analysis for iterative prompting, and the utility of tests in terms of evaluating and improving code optimization. We would be happy to address any further questions you may have.

---

### Official Review · Reviewer_HVF9 · 2025-07-01

**Clarity:** 3
**Significance:** 2
**Originality:** 3
**Rating:** 5
**Confidence:** 4

**Summary:**

This paper proposes a framework, WEDGE, for generating additional inputs for performance tests based on the performance profiles of already available inputs. The framework leverages LLM for synthesizing performance constraints from slow and fast execution profiles from already available inputs. These constraints are then integrated with a fuzzer to generate performance-stressing test inputs. Experimental results show that WEDGE generated tests lead programs to execute, on average, 84.5%
more CPU instructions than the SOTA approach.

**Questions:**

1. Utility experiments with EFFI-LEARNER use top-5 slowest inputs from less than ten available tests. How does it compare to fuzzing without constraint generation?

2. Given a performance constraint can be synthesized, can LLM improve the code given the performance constraints as prompt (without generating additional inputs)?

**Ethical Concerns:**

["NO or VERY MINOR ethics concerns only"]

**Final Justification:**

This paper presents a new approach of generating performance constraints that addresses a timely challenge of generating high-performance code using LLM. The authors provide a clear description of the approach and experimental results show the effectiveness of the proposed approach. The ablation studies strongly justify the contributions of this work.

**Limitations:**

Yes.

**Paper Formatting Concerns:**

None.

**Quality:**

3

**Strengths And Weaknesses:**

Strengths

- The paper targets an challenging problem of evaluating performance for efficient LLM code generation. Overall, the paper is a well written and the proposed approach is easy to follow.

- Leveraging LLM to synthesize performance constraints and use that as an intermediate step to generate more targeted performance-stressing input is a novel approach. Experimental results show utility of the proposed approach in improving performance of LLM generated code.


Weakness

- The improvements are measured using CPU instructions which may not directly translate to overall performance improvement of a program (for example, if the program bottleneck is memory usage). It would be interesting to how the results compare to execution time based performance numbers.

- The proposed method relies on availability of slow and fast inputs but the baseline used for utility experiments (EFFIE-LEARNER and PIE) do not rely on such constraints. Although the author clearly expressed this limitation, I think this dependency significantly reduces practical utility of the proposed approach.

---

> ### Author Rebuttal · Authors · 2025-07-31
>
> We really appreciate your time and effort in reviewing our paper and giving us constructive comments. We address your comments and questions in the following.
>
> **Q1: The proposed method relies on availability of slow and fast inputs but the baseline used for utility experiments (EFFIE-LEARNER and PIE) do not rely on such constraints. Although the author clearly expressed this limitation, I think this dependency significantly reduces practical utility of the proposed approach.**
>
> We agree that the dependency on "fast/slow" inputs needs clearer explanation and will improve this part in the next iteration of the paper. WEDGE mines contrastive pairs once per program from whatever tests are already available and come with the target (e.g., unit tests). If the target program comes with no tests, one can bootstrap it by generating valid test/inputs using a fuzzer. In both cases, such contrastive inputs do not necessarily have to be performance-stressing, as long as they provide diagnostic hints to help LLMs to quickly localize the program points to stress. In the worst case, this independent stage can be as easy as a simple prompt, which makes it more practical, or it can rely on specialized profilers for a more accurate diagnosis. In fact, our evaluation simply leverages CodeContests tests for correctness evaluation, not performance.
>
> **Q2: Utility experiments with EFFI-LEARNER use top-5 slowest inputs from less than ten available tests. How does it compare to fuzzing without constraint generation?**
>
> Good question. Following the setting in our utility experiment in Table 2, we added the experiment by using the top-5 slowest inputs generated by plain AFL++, and integrated these tests into EFFI-LEARNER to measure the improvement it brought. Due to the limited time frame, we subsample 14 problems for this experiment.
>
> | Profiling           | Execution Time | Total Memory Usage | Max Memory Usage | Execution Time (ict) |
> |---------------------|----------------|---------------------|------------------|-----------------------|
> | Wedge tests       | 45.33%     | 56.26%          | 38.86%       | 73.98%            |
> | AFL++ tests        | 14.80%         | 18.69%              | 11.84%           | 26.07%                |
>
> Using AFL++ tests’ execution profile as the prompt leads to substantially decreased optimization performance. Wedge’s tests lead to the optimized program to execute 45.33% fewer instructions compared to the original program, while AFL++ only achieves 14.8%. We attribute this observation to the Wedge’s generated tests being more performance-characterizing than those of AFL++. We are working on scaling the experiment to the full set and repeating the running 5 times to mitigate randomness.
>
> **Q3: Given a performance constraint can be synthesized, can LLM improve the code given the performance constraints as prompt (without generating additional inputs)?**
>
> This is a very interesting suggestion. Conceptually, PC constraints alone can already characterize inefficient implementation (e.g., data structure access time complexity), though not as concrete as execution feedback. However, it is unclear how LLMs consume such information when tasked to optimize the code. To this end, we conduct a few case studies using the samples instrumented with interesting and informative constraints (see Appendix in the paper).
>
> Specifically, we used the 5 programs from our appendix that discuss the case studies, where the PC-constraints (pcc) are readily available for reviewers to check in the appendix. As a baseline method, we asked LLM to optimize the original program, without providing PC-constraints (LLM optimized no pcc). We then asked LLM (in a new chat) to optimize the original program while providing pc-constraints (LLM optimized w/ pcc). We made the following observations.
>
> Observation #1: Mixed results, in 3/5 cases, with or without PC-constraints, the program is optimized in a similar way, so PC-constraints don't help or hurt.
>
> Observation #2: In 1/5 cases, PC-constraints appear to hurt the optimization, but a more in-depth analysis shows that the LLM using PC-constraints actually optimizes the program in a lower/better complexity class, but since the parameters are small, the "less optimized" version runs faster on the "average" case.
>
> Observation #3: in 1/5 cases, both optimized versions have logic bugs in how they optimize and re-implement the original algorithm; This is the largest program provided to LLM out of all 5 (~100 vs ~50 LoCs)
>
> | technique              | 16B-sol-34   | 546C-sol-567 | 633A-sol-622  | 1209B-sol-328 | 1182E-sol-22 |
> |------------------------|-------------:|-------------:|--------------:|--------------:|-------------:|
> | **Average instructions count** |             |             |              |              |             |
> | Original               | 1000323782.5 |   19583381.3 |   129818315.6 |    23436043.2 | n/a  |
> | LLM optimized no pcc   |    2261541.6 |    2336769.4 |      140243.7 |     2920159.2 | n/a |
> | LLM optimized w/ pcc   |    2261414.3 |    2320602.1 |      139907.5 |     5337538.5 | n/a |
> | **Median instructions count** |         |             |              |              |             |
> | Original               | 1596548458.0 |    2225882.0 |     2927145.5 |    30180433.0 | n/a |
> | LLM optimized no pcc   |    2274604.0 |    2297157.0 |      139920.0 |     3207311.0 | n/a |
> | LLM optimized w/ pcc   |    2274475.5 |    2281172.0 |      139855.0 |     6327841.5 | n/a |
>
> Our preliminary case study cannot confirm that using the performance constraints directly as a prompt can improve code optimization, as opposed to the execution feedback, as shown in our experiments for Q2.
>
> **Q4: The improvements are measured using CPU instructions which may not directly translate to overall performance improvement of a program (for example, if the program bottleneck is memory usage). It would be interesting to how the results compare to execution time based performance numbers.**
>
> Due to space limitations, we reported physical running times for our experiments in Appendix B.1. We will label them more clearly and add an explicit forward reference in the main body of the paper. These results are summarized in Table 4, and the trends are consistent with the instruction count results in Table 1. We use instruction counts as the primary metric because it has a much lower variance, as shown by prior work [1] and by our own measurements. Our experiments observe ~0.03% variance for instructions versus ~12% for physical running time, which can be influenced by I/O, background tasks, and scheduling noise, particularly for I/O‑bound programs like ours.
>
> WEDGE is agnostic to the metrics used. Any per-input cost calculation can be used to inform contrastive pair mining and the fuzzer's input mutation engine. For example, if memory behavior is the bottleneck, the instruction counts can be replaced with metrics like peak memory usage, heap bytes allocated, allocation count, cache‑miss events, or garbage-collection time without changing the workflow of our tool. However, we do note that adopting those types of metrics may require additional instrumentation or specialized tools like memory profilers. Also, note that the metric itself only affects the running time of the programs under test, while everything else (prompting, PC-constraint generation, instrumentation) remains unaffected.
>
> [1] Liu, Jiawei, Songrun Xie, Junhao Wang, Yuxiang Wei, Yifeng Ding, and Lingming Zhang. "Evaluating language models for efficient code generation." arXiv preprint arXiv:2408.06450 (2024).

---

### Official Review · Reviewer_b8hS · 2025-07-01

**Clarity:** 3
**Significance:** 3
**Originality:** 3
**Rating:** 5
**Confidence:** 4

**Summary:**

The paper presents WEDGE, a novel framework that synthesizes performance-characterizing constraints to generate performance-stressing test inputs for code. By leveraging contrastive execution profiling and guiding fuzzing via LLM-inferred constraints, WEDGE exposes inefficiencies that size-based test generation methods often miss. The resulting test suite, PERFFORGE, stresses programs more effectively, leading to better optimization and evaluation outcomes when used with state-of-the-art tools like EFFI-LEARNER and PIE. The authors support their claims with robust empirical results and thoughtful ablation studies, making this work a compelling contribution to the field of LLM-based code evaluation and optimization.

**Questions:**

* What is the cost in time and tokens?
* I'd suggest using an anonymous github https://anonymous.4open.science/ instead.
* Given the iterative prompt-refine loop and profiling overhead, can you quantify the average cost (in tokens/runtime) per test case?
* Do different constraints generated for the same problem typically yield different slow paths? Or is there convergence to a few common patterns?

**Ethical Concerns:**

["NO or VERY MINOR ethics concerns only"]

**Final Justification:**

Thank you. I'll keep my score. The author successfully resolved my concerns about the generalizability. However, I still feel that the each solution requires profiling and prompting, which is something I don't find the best option.

**Limitations:**

* The paper is somewhat narrowed down to a specific domain and the evaluation is focused on competitive programming-style problems.
* The authors do not explore the generalizability to real-world systems or codebases is not explored.

**Paper Formatting Concerns:**

Nothing

**Quality:**

3

**Strengths And Weaknesses:**

Strengths:
* One thing that I like is that WEDGE avoids naive input size scaling by synthesizing performance-characterizing constraints, like gcd(a, b) not dividing c in a nested loop problem. I feel that this enables precise targeting of inefficiencies beyond just large inputs, for instance, for the Diophantine example in Figure 2.
* Contrastive profiling and fuzzing seems quite novel and effective.
* Good evaluation results
* Strong ablation study and the results justify the proposed solution.

Weaknesses:
* I don't like that each solution requires profiling, prompting and instrumentation.
* The paper is somewhat narrowed down to a specific domain and the evaluation is focused on competitive programming-style problems.
* The authors do not explore the generalizability to real-world systems or codebases is not explored.

---

> ### Author Rebuttal · Authors · 2025-07-31
>
> We thank the reviewer for your time and effort in reviewing our paper and providing constructive comments! We have used an anonymized GitHub to link to our code: https://anonymous.4open.science/r/wedge-617E/README.md
>
> **Q1: What is the cost in time and tokens? Quantify the average cost (in tokens/runtime) per test case**
>
> Thanks for pointing out this important question. We break the cost down for different stages as follows (**per solution program**):
>
> | Stage   | Tokens | Time |
> | -------- | ------- | ------- |
> | contrastive input pair mining | N/A  |  99.65 executions, 15.46s  |
> | profiling feedback collection | N/A   | 2 executions |
> | PC constraint reasoning and constraint checker generation  |  9633.16  |  N/A  |
> | constraint-aware mutator generation   |  6905.27  |  180s fuzzing dry run  |
> | constraint-guided fuzzing   |  N/A  | 1 hour fixed budget   |
> **Average test cases generated: 63.99.**
>
> We will include the results in the updated version of the paper.
>
>
> **Q2: The paper is somewhat narrowed down to a specific domain and the evaluation is focused on competitive programming-style problems**
>
> Thanks for pointing this out. Our key motivation stems from the observation that the emerging real-world use of LLMs for optimizing code efficiency [1,2] does not match the existing evaluation practices (largely based on correctness tests or length-stressing tests).
>
> **Narrowed down to a specific domain**. In terms of the scope, we observe that LLMs for code optimization have been related to many recent seminal efforts [3,4], let alone the significant body of research papers on LLM for compiler optimization. However, we understand this can be subjective, and there is a great gap between their testing requirements. We will add the discussion correspondingly in the paper.
>
> **Focus on competitive programming-style problems**. We absolutely agree that moving beyond competitive programming-style problems for evaluation is exciting [4]. We cautiously believe that the unique design in Wedge, which decouples global test input generation from local performance bottleneck reasoning, makes it a promising solution for project-level performance testing. Recall that PC-constraints are small predicates over local program states that, when true, tend to indicate the code reaches an inefficient state (e.g., "array nearly sorted so partitioning degenerates", "graph has high-degree nodes", "regex causes computationally expensive backtracking"). If we move from competitive programming-style problems to, say, a compression library (which uses sort and graph-like data structures), Wedge’s decoupled workflow remains the same: synthesize a local predicate, then guide a fuzzer to generate inputs towards satisfying it. We will add studies to extend Wedge for project-level performance testing in the next version of the paper.
>
> [1] Rina Diane Caballar. “Python Profiler Links to AI to Improve Code Scalene identifies inefficiencies and asks GPT-4 for suggestions”, https://spectrum.ieee.org/python-programming
>
> [2] Lin, Hannah, Martin Maas, Maximilian Roquemore, Arman Hasanzadeh, Fred Lewis, Yusuf Simonson, Tzu-Wei Yang et al. "ECO: An LLM-driven efficient code optimizer for warehouse scale computers." arXiv preprint arXiv:2503.15669 (2025).
>
> [3] Novikov, Alexander, Ngân Vũ, Marvin Eisenberger, Emilien Dupont, Po-Sen Huang, Adam Zsolt Wagner, Sergey Shirobokov et al. "AlphaEvolve: A coding agent for scientific and algorithmic discovery." arXiv preprint arXiv:2506.13131 (2025).
>
> [4] Ouyang, Anne, Simon Guo, Simran Arora, Alex L. Zhang, William Hu, Christopher Ré, and Azalia Mirhoseini. "Kernelbench: Can llms write efficient gpu kernels?." arXiv preprint arXiv:2502.10517 (2025).
>
> **Q3: The authors do not explore the generalizability to real-world systems or codebases is not explored.**
>
> We acknowledge that our experiments in the current version do not cover large-scale codebases. However, we believe a few straightforward changes can enable our tool to handle such scale:
>
>     1. The AFL++ fuzzer in Wedge can be swapped for a fuzzer that operates at the function/API‑level [1,2,3] or computes a slice around a target function [4].
>
>     2. Wedge would synthesize PC‑constraints for each target function and instrument the function following our original setup.
>
>     3. Wedge additionally sets the checker as the coverage objective [5] or relies on path pruning [6,7] to remove unreachable paths.
>
>     4. WEDGE currently uses end-to-end performance counters as a metric, but this can be changed with more sophisticated objectives, such as a multi-dimensional vector of execution counts for basic blocks [8].
>
> [1] Babić, Domagoj, Stefan Bucur, Yaohui Chen, Franjo Ivančić, Tim King, Markus Kusano, Caroline Lemieux, László Szekeres, and Wei Wang. "Fudge: fuzz driver generation at scale." In Proceedings of the 2019 27th ACM Joint Meeting on European Software Engineering Conference and Symposium on the Foundations of Software Engineering, pp. 975-985. 2019.
>
> [2] Sherman, Gabriel, and Stefan Nagy. "No Harness, No Problem: Oracle-guided Harnessing for Auto-generating C API Fuzzing Harnesses." In 2025 IEEE/ACM 47th International Conference on Software Engineering (ICSE), pp. 775-775. IEEE Computer Society, 2025.
>
> [3] Serebryany, Kostya. "{OSS-Fuzz}-Google's continuous fuzzing service for open source software." (2017).
>
> [4] Murali, Aniruddhan, Noble Mathews, Mahmoud Alfadel, Meiyappan Nagappan, and Meng Xu. "FuzzSlice: Pruning false positives in static analysis warnings through function-level fuzzing." In Proceedings of the 46th IEEE/ACM International Conference on Software Engineering, pp. 1-13. 2024.
>
> [5] Böhme, Marcel, Van-Thuan Pham, Manh-Dung Nguyen, and Abhik Roychoudhury. "Directed greybox fuzzing." In Proceedings of the 2017 ACM SIGSAC conference on computer and communications security, pp. 2329-2344. 2017.
>
> [6] Srivastava, Prashast, Stefan Nagy, Matthew Hicks, Antonio Bianchi, and Mathias Payer. "One fuzz doesn’t fit all: Optimizing directed fuzzing via target-tailored program state restriction." In Proceedings of the 38th Annual Computer Security Applications Conference, pp. 388-399. 2022.
>
> [7] Huang, Heqing, Yiyuan Guo, Qingkai Shi, Peisen Yao, Rongxin Wu, and Charles Zhang. "Beacon: Directed grey-box fuzzing with provable path pruning." In 2022 IEEE Symposium on Security and Privacy (SP), pp. 36-50. IEEE, 2022.
>
> **Q4: I don't like that each solution requires profiling, prompting and instrumentation.**
>
> Thank you for raising this concern about the overhead. We wish to clarify that most of these costs are incurred only once per target program in an offline setup phase. The remaining costs are limited to small compile‑time overhead when instrumenting the target programs with synthesized checkers. To collect execution profiles and mine contrasting input pairs, we only need to run the program on a predefined set of tests once at the beginning. This is an offline, per‑target step that produces reusable profiles across fuzzing campaigns and LLM model iterations. Similarly, Wedge prompts the LLM once per target program, at the beginning, to synthesize PC‑constraints, its checkers, and the custom, constraint-aware mutators, and instrument the target programs. All subsequent runs and fuzzing campaigns will amortize the costs. Considering the generated tests will be fixed for future use, we believe the one-time cost of these steps is reasonable. We will add the discussion in the next version of the paper.

---

> > ### Comment · Reviewer_b8hS · 2025-08-07
> >
> > Dear authors,
> >
> > Thank you for your comment. I have no further concerns about the paper.
> >
> > Best regards

---

> > > ### Author Response · Authors · 2025-08-09
> > >
> > > Dear reviewer b8hS,
> > >
> > > We sincerely thank you for acknowledging our rebuttal and the valuable comments. Your insightful questions indeed improve our paper, as acknowledged by another reviewer, zqXK, as well. And we appreciate you taking the time to engage with our rebuttal.
> > >
> > > Best regards

---

### Official Review · Reviewer_trS2 · 2025-07-04

**Clarity:** 3
**Significance:** 3
**Originality:** 3
**Rating:** 5
**Confidence:** 4

**Summary:**

This paper proposes WEDGE, a framework for generating performance-stressing tests given the program under test.

WEDGE uses a combination of LLMs and fuzzers to generate such tests. It first collects some contrasting pairs of test inputs that are similar on the surface but differs in instruction count to run. Then it utilizes LLMs to analyze the performance critical regions in the code and generate performance-characterizing constraints at the beginning of these regions. After the constraints are inserted, a fuzzer is used to conduct coverage-guided fuzzing, which needs to get through these constraints to improve coverage, thus satisfying these constraints.

Experiments show that WEDGE-generated tests can greatly slow the programs down. Also in terms of downstream tasks, WEDGE tests can be a better feedback signal for performance-improving refinements.

**Questions:**

1. How generalizable is this method to more complicated projects compared to single programs?

**Ethical Concerns:**

["NO or VERY MINOR ethics concerns only"]

**Final Justification:**

I'm keeping my (already-high) rating.

**Limitations:**

yes

**Quality:**

3

**Strengths And Weaknesses:**

### Strengths:

1. The combination of LLMs and fuzzing is interesting and intuitive. LLMs, with their knowledge about algorithms and data structures and their worst-case scenarios, are used to identify the performance-critical parts and generate constraints on performance-stressing program states. While fuzzers are used to actually create inputs that meet these constraints.

2. Very clear advantage over baselines and significant improvements in downstream tasks. The method could have impacts on test-time scaling of compute for coding LLMs as better signals can improve iterative refinement. It may also serve as better rewards for model training.

---

> ### Author Rebuttal · Authors · 2025-07-31
>
> We are grateful for your time, effort, and leaving encouraging comments. We address your comments and questions in the following.
>
> **Q1: How generalizable is this method to more complicated projects compared to single programs?**
>
> We do not attempt to overclaim Wedge can generalize to complicated projects without empirical evidence. However, Wedge’s reliance on fuzzing that readily supports large projects and its design in constraining LLMs for local performance constraint reasoning makes it promising to generalize to larger projects. Specifically, Wedge can be applied to generate performance constraints at the local function- or slice-level [1] of the project. Such a local reasoning avoids relying on LLM’s limited capabilities in long-context reasoning [2, 3, 4, 5], while fuzzers bridge such gaps to enable global input search for both whole-project or library testing [6]. In addition, optimizing at the local code context can still benefit the entire project [7, 8, 9].
>
> Our main experiments are motivated by the emerging attempt to use LLMs for optimizing code efficiency based on existing code benchmarks. Their evaluation, however, is largely based on the benchmark’s existing correctness tests or length-stressing tests. We totally agree that project-level performance evaluation is exciting and can require more agentic design [4]. We will add studies that extend Wedge for project-level performance testing in the updated version of the paper. We appreciate the comments, which give us the opportunity to further clarify the advantages of Wedge’s design.
>
> [1] Murali, Aniruddhan, Noble Mathews, Mahmoud Alfadel, Meiyappan Nagappan, and Meng Xu. "FuzzSlice: Pruning false positives in static analysis warnings through function-level fuzzing." In Proceedings of the 46th IEEE/ACM International Conference on Software Engineering, pp. 1-13. 2024.
>
> [2] Liu, Tianyang, Canwen Xu, and Julian McAuley. "Repobench: Benchmarking repository-level code auto-completion systems." arXiv preprint arXiv:2306.03091 (2023).
>
> [3] Jimenez, Carlos E., John Yang, Alexander Wettig, Shunyu Yao, Kexin Pei, Ofir Press, and Karthik Narasimhan. "Swe-bench: Can language models resolve real-world github issues?." arXiv preprint arXiv:2310.06770 (2023).
>
> [4] Shetty, Manish, Naman Jain, Jinjian Liu, Vijay Kethanaboyina, Koushik Sen, and Ion Stoica. "GSO: Challenging Software Optimization Tasks for Evaluating SWE-Agents." arXiv preprint arXiv:2505.23671 (2025).
>
> [5] He, Xinyi, Qian Liu, Mingzhe Du, Lin Yan, Zhijie Fan, Yiming Huang, Zejian Yuan, and Zejun Ma. "SWE-Perf: Can Language Models Optimize Code Performance on Real-World Repositories?." arXiv preprint arXiv:2507.12415 (2025).
>
> [6] Serebryany, Kostya. "{OSS-Fuzz}-Google's continuous fuzzing service for open source software." (2017).
>
> [7] Rina Diane Caballar. “Python Profiler Links to AI to Improve Code Scalene identifies inefficiencies and asks GPT-4 for suggestions”, https://spectrum.ieee.org/python-programming
>
> [8] Bala, Vasanth, Evelyn Duesterwald, and Sanjeev Banerjia. "Dynamo: A transparent dynamic optimization system." In Proceedings of the ACM SIGPLAN 2000 conference on Programming language design and implementation, pp. 1-12. 2000.
>
> [9] Sasnauskas, Raimondas, Yang Chen, Peter Collingbourne, Jeroen Ketema, Gratian Lup, Jubi Taneja, and John Regehr. "Souper: A synthesizing superoptimizer." arXiv preprint arXiv:1711.04422 (2017).

---

> > ### Comment · Reviewer_trS2 · 2025-08-06
> >
> > Thanks for the response. Excited to see WEDGE extended to more complicated projects in the future.

---

> > > ### Author Response · Authors · 2025-08-06
> > >
> > > We appreciate your encouraging response. We are also very excited about this research direction and look forward to sharing our results on more complex projects. Thank you again for your valuable feedback.

---

### Author Response · Authors · 2025-08-09
**Response Summary**

We really appreciate your efforts in overseeing the review process for our submission. We understand you have limited time to review each submission, so we summarize our work, rebuttals, and discussions briefly as follows:

Wedge aims to evaluate the existing LLM-based code efficiency optimization techniques by generating more performance-stressing tests. Existing approaches either focus on using correctness tests or leverage LLMs to directly generate the inputs, which is too challenging that the generated inputs often reduce to length-stressing. Wedge’s key contribution is to decompose the problem of stress test generation into local performance-characterizing constraint reasoning and global constraint-guided search. The former alleviates the reasoning burden of LLMs to only the local code context, while the latter can leverage efficient input search tools based on fuzzing. We demonstrate that Wedge substantially outperforms the state-of-the-art approaches. We also demonstrate that the generated tests facilitate more effective evaluation and improvement of LLM-based code optimization approaches.

In response to the reviewers’ questions, we have added the following new experiments and clarifications. As acknowledged by the reviewers, our responses and additional experiments have addressed all the reviewers’ questions.

* We added the experiment detailing the cost in time and tokens per solution program for each stage of Wedge (reviewer `b8hS`), and demonstrated that the estimated cost per test case is 258 tokens and 59s.

* We clarified that Wedge’s reliance on "fast/slow" contrastive inputs is not a limitation. They can be easily obtained from available tests that are not necessarily performance-stressing, and can also be sampled using a fuzzer (reviewer `HVF9`).

* We have clarified that Wedge’s generic design can generalize to complex projects in principle, and outlined how to extend Wedge for project-level performance testing (reviewer `trS2`, `b8hS`, `zqXK`).

* We added a new preliminary experiment on the existing performance optimization baseline (EFFI-Learner) by prompting with the tests generated by the baseline fuzzer (AFL++) and Wedge. We show that Wedge’s tests enable significantly improved optimization effectiveness (reviewer `HVF9`), ranging from 27.02 percentage points (max memory usage) to 47.91 percentage points (#instructions).

* We added a case study that tasks LLM to optimize code w/ and w/o Wedge’s PC-constraints. We observe mixed results, indicating that using the constraints directly as a prompt does not significantly improve code efficiency compared to generating tests and relying on the corresponding execution feedback (reviewer `HVF9`).

* We clarified that physical running times are reported in Appendix B.1 and are consistent with instruction counts, and that Wedge is agnostic to the metrics used (reviewer `HVF9`).

* We added an extensive discussion of performance-fuzzing-related works with new experiments comparing Wedge's efficiency and performance-stressing capabilities against PerfFuzz (reviewer `zqXK`). Our results show Wedge can generate more stressing tests, outperforming PerfFuzz by 2.4x in #instructions (reviewer `zqXK`).

* We added measurement of the fuzzing speed and showed that AFL with PerfFuzz’s profiling incurs more overhead than fuzzing with our instrumented programs (31.49% slower than Wedge). However, the increased profiling cost does not lead to improved performance-stressing capabilities. This supports our claim regarding the profiling overhead of traditional performance fuzzing approaches (reviewer `zqXK`).

* We clarified that Wedge is primarily for generating performance tests to benchmark code optimization techniques, not a bug-finding tool, and currently does not support stateful fuzzing (reviewer `zqXK`).
We sincerely appreciate the significant effort you put into the review process. We are grateful to all the reviewers whose comments significantly helped improve our paper.

We sincerely appreciate the significant effort you put into the review process. We are extremely grateful for the valuable questions, feedback, and comments from the reviewers. Our paper is indeed improved with the help of reviewers.

---

### Note · Authors · 2025-08-16

Thank you for overseeing the review process. We appreciate the valuable feedback from all the reviewers that helped strengthen our paper. Besides the response summary we provided earlier, we hope to update our additional evaluation against the performance fuzzing baseline, PerfFuzz (and its variant), thanks to the constructive suggestion by the reviewer `zqXK`. Our last response has reported the results on 53% of our evaluation set. Today, we have finally got the complete results for our full evaluation dataset. We include them below (corresponding to Table 1 in our paper).

Instruction count (×$10^8$):
| Technique | Average | Median | Win rate (%) |
| :--- | :--- | :--- | :--- |
| Wedge | 5.83 | 0.82 | 60% |
| PerfFuzz (original programs) | 2.96 (↓2.0×) | 0.51 (↓1.7×) | 7% |
| PerfFuzz (instrumented programs) | 2.95 (↓1.9×) | 0.51 (↓1.7×) | 6% |
| TG-prompt | 3.75 (↓1.6×) | 0.66 (↓1.3×) | 13% |
| EvalPerf_slow | 2.87 (↓2.0×) | 0.46 (↓1.8×) | 8% |
| EvalPerf_rand | 2.74 (↓2.1×) | 0.48 (↓1.7×) | 6% |


Physical execution time (ms):
| Technique | Average | Median | Win rate (%) |
| :--- | :--- | :--- | :--- |
| Wedge | 217.86 | 123.96 | 60% |
| PerfFuzz (original programs) | 101.65 (↓2.1×) | 56.67 (↓2.2×) | 7% |
| PerfFuzz (instrumented programs) | 100.56 (↓2.2×) | 55.52 (↓2.3×) | 6% |
| TG-prompt | 179.32 (↓1.3×) | 89.35 (↓1.4×) | 13% |
| EvalPerf_slow | 133.81 (↓1.6×) | 61.89 (↓1.6×) | 8% |
| EvalPerf_rand | 126.56 (↓1.7×) | 61.02 (↓1.6×) | 6% |

The extended results are consistent with the prior smaller-scale experiments, demonstrating Wedge can substantially outperform PerfFuzz (and its variant). Specifically, it generates tests 2.0X (#instructions) / 2.1X (running time) slower than PerfFuzz, similar to our previous observations and findings reported in the last updates. On 60% solution programs, Wedge can generate the most stressing tests, outperforming PerfFuzz with a win rate of 7%.

---

### Decision · Program_Chairs · 2025-09-17

**Decision:**

Accept (poster)

**Comment:**

This paper addresses the interesting problem of generating performance-stressing test inputs for evaluating LLM-based code optimization techniques. The work tackles a timely challenge where current evaluation practices rely on correctness tests or simple length-stressing inputs that fail to reveal nuanced performance bottlenecks. The proposed WEDGE framework presents a novel approach that combines LLM-synthesized performance-characterizing constraints with fuzzing techniques to generate more effective performance tests.

The reviewers consistently praised the paper's technical contributions, highlighting the intuitive combination of LLMs with fuzzing, significant empirical improvements over baselines, and strong ablation studies. They also appreciated the clear presentation and demonstrated utility for improving code optimization effectiveness. On the other hand, reviewers raised concerns about the work's scope being limited to competitive programming problems rather than real-world systems, the overhead of requiring profiling and instrumentation for each solution, and the dependency on contrasting fast/slow input pairs. One reviewer also noted issues with the characterization of related work and missing comparisons with established performance fuzzing techniques.

The authors comprehensively addressed these concerns in their rebuttal through additional experiments, cost analyses, comparisons with performance fuzzing baselines, and honest discussions of current limitations.